# Hybrid computational modeling demonstrates the utility of simulating complex cellular networks in type 1 diabetes

**Zhenzhen Shi**[1,2], **Yang Li**[3], **Majid Jaberi-Douraki**[1,2]*

**1** 1DATA Consortium, Kansas State University Olathe, Olathe, Kansas, United States of America,
**2** Department of Mathematics, Kansas State University, Manhattan, Kansas, United States of America,
**3** Laboratory of Immunology and Nanomedicine, Shenzhen Institute of Advanced Technology (SIAT), Chinese Academy of Science, Shenzhen, China

* jaberi@k-state.edu

**Data Availability Statement:** Source code is available at https://github.com/szztracy/T1D-Simulation.

## Abstract

Persistent destruction of pancreatic β-cells in type 1 diabetes (T1D) results from multifaceted pancreatic cellular interactions in various phase progressions. Owing to the inherent heterogeneity of coupled nonlinear systems, computational modeling based on T1D etiology help achieve a systematic understanding of biological processes and T1D health outcomes. The main challenge is to design such a reliable framework to analyze the highly orchestrated biology of T1D based on the knowledge of cellular networks and biological parameters. We constructed a novel hybrid in-silico computational model to unravel T1D onset, progression, and prevention in a non-obese-diabetic mouse model. The computational approach that integrates mathematical modeling, agent-based modeling, and advanced statistical methods allows for modeling key biological parameters and time-dependent spatial networks of cell behaviors. By integrating interactions between multiple cell types, model results captured the individual-specific dynamics of T1D progression and were validated against experimental data for the number of infiltrating CD8+T-cells. Our simulation results uncovered the correlation between five auto-destructive mechanisms identifying a combination of potential therapeutic strategies: the average lifespan of cytotoxic CD8+T-cells in islets; the initial number of apoptotic β-cells; recruitment rate of dendritic-cells (DCs); binding sites on DCs for naïve CD8+T-cells; and time required for DCs movement. Results from therapy-directed simulations further suggest the efficacy of proposed therapeutic strategies depends upon the type and time of administering therapy interventions and the administered amount of therapeutic dose. Our findings show modeling immunogenicity that underlies autoimmune T1D and identifying autoantigens that serve as potential biomarkers are two pressing parameters to predict disease onset and progression.

## Author summary

Pancreatic β-cells secreting insulin upon metabolic demand are the core to regulate glucose concentration for healthy states. Type 1 diabetes (T1D) is a group of metabolic

**Funding:** MJD received funding from BioNexus KC (https://bionexuskc.org/). YL received funding from National Natural Science Foundation of China (31701005). The funders had no role in study design, data collection and analysis, decision to publish, or preparation of the manuscript.

**Competing interests:** The authors have declared that no competing interests exist.

disorders in which β-cells are targeted by biased decisions of the immune system. The barrier in sampling experimental data from pancreatic tissues in high-risk T1D subjects and the complexity of the mechanisms regulating T1D makes the use of quantitative modeling approaches an intriguing opportunity to analyze this disease. To decipher complex system behaviors resulting from inter- and intra-cellular and signaling networks linking the immune system and metabolism during T1D progression, we developed a hybrid computational framework to unravel high computational complexity levels for individual-specific T1D development in non-obese diabetic mice. We also discovered non-intuitive biological parameters, the potential for therapeutic strategies. These effective strategies associated with hybrid agent-based modeling can serve as a prototype for *in- silico* experiments of therapy-directed trials potentially regulating T1D progression. We expect that the proposed methods will play an important role in the quest for standardized applications of agent-based modeling to simulating complex disease progression.

## Introduction

Various autoimmune disorders influence human health; type 1 diabetes (T1D), a form of diabetes mellitus in humans and animal research, is a group of metabolic disorders in which insulin-secreting β-cells are targeted by biased decisions of the immune system. T1D progression after initiation follows multiple phase transitions in complex pancreatic cellular networks. As the primary infiltration of the immune response against self-antigens proceeds, numerous interactions including cell differentiation and competition between different types of cells accelerate the induction of autoimmune responses, which eventually leads to T1D onset and progression.

The barrier in sampling experimental data from pancreatic tissues or lymph nodes in high-risk T1D subjects makes the use of mathematical/computational modeling of pancreatic β-cell destruction an intriguing opportunity to analyze this disease. Various questions and problems typically arise from researchers and the disease on a daily basis that require long and tedious experimental work to possibly find an answer. Since the model has been trained on multiple datasets, having such a practical and flexible platform can widen the horizon with reasonable accuracy and provide preliminary insight into the appeal of such questions as the ABM model can be quickly adjusted to address such inquires and issues before conducting any experiments. In addition, we can all acknowledge designing an experiment requires lengthy paperwork and follows with associated costs assuming all the tools, materials, patients are available for lab work. Moreover, experiments alone cannot often explain the behavior of very rich and complex developmental dynamics of pancreatic islets and β-cells with feedback across different levels of biological organization as pointed out in Anmar and Santiago's review [1]. Such a quantitative approach emerging from modeling the detailed biology of immune responses could provide novel insights into the mechanisms underlying the regulation of T1D when the safety, reproducibility, and efficiency of the present experimental techniques are challenging.

Now the challenge is to design and construct such a quantitative framework for analyzing highly orchestrated biology, immunology, and the pathogenesis of T1D based on the current knowledge and modular organization of cellular networks and biological parameters. To unravel complex system behaviors resulting from the inter- and intra-cellular and signaling networks linking the immune system and metabolism during T1D progression, it is crucial to first integrate essential information from inherent biological processes into a systematic framework.

Mathematical models have been widely employed to study systematic behaviors in disease progression [2–8]. Our previous studies focused on the mathematical modeling of T1D progression using novel object-oriented approaches such as integro-differential equation modeling with reasonably defined sets of criteria that determined susceptibility to T1D [9–12]. Compared to mathematical models, agent-based modeling (ABM) has been used to model cell heterogeneity from single-cell variations to mixed cells by simulating the behaviors of individual agents [13–17]. Depending on the behaviors of individual agents in a biological system, agents interact with each other and move over a simulation environment following a set of pre-defined rules [18].

Linderman et al. applied an agent-based approach to examine the relationships between parameters related to dendritic cell-T cell interactions and the output of primed T cells [19]. There are also a number of agent-based models focusing on different aspects of T1D progression. Wedgwood et al. developed an agent-based model to investigate the progression of insulitis in human T1D by simulating interactions among B cells, T cells, and beta cells [20]. Additionally, Ozturk et al. proposed an agent-based model focusing on modeling interaction between beta cells and different types of CD8$^+$ T cells [21]. Later, Xu et al. proposed another agent-based model to simulate the effect of regulatory T cells on T1D progression [22]. Compared to the previous studies, we focus on modeling the detailed biology of dendritic cells and their interactions with CD8$^+$ T cells, beta cells, and antigens. Moreover, thorough and expensive statistical methods were utilized to analyze the outputs of the model and provided a solid foundation for therapy-directed simulations. This model was also designed to investigate individual-specific dynamics of CD8$^+$ T cells which were observed in NOD mouse experiments.

The non-obese diabetic (NOD) mouse is a well-established animal model that exhibits striking similarities to the disease in humans. NOD mouse models sharing a number of genetic and immunologic traits with human T1D [23–25] serve as testing arenas for potential treatments and reliable and prominent tools for studying the pathogenesis of T1D in humans [26]. Since identifying safe methods for sampling and visualizing the human endocrine pancreas remains incompletely understood [27], autoimmunity during T1D progression has been investigated extensively in NOD mouse models [23,24]. Therefore, an abundance of data is accessible for accurate incorporation of relationships among agents to support the establishment of ABM.

Based on the above facts, we proposed a novel hybrid agent-based model by integrating phase progressions in complex pancreatic cellular networks, mathematical modeling, biological parameters, and ABM into a computational framework to simulate T1D progression using experimental data and clinical evidence from NOD mouse models. Through exploiting ABM, the aim of this study includes the following four aspects: (1) to develop a hybrid model based upon the underlying mechanisms summarized from the NOD mouse model and then cross-validate the model with experimental data; (2) to predict pancreatic β cell destruction targeted by the immune system during T1D progression; (3) to identify key parameters in driving autoimmunity and estimate effects of these parameters on determining how sensitive (or uncertain) different mechanisms of the model are to perturbations and thus identifying potential targets for therapeutic purposes; and (4) to conduct therapy-directed simulations and explore immune-intervention strategies for predicting the onset and progression of the disease and potentially treating T1D.

## Materials and methods

### Abstraction of modeling performance

This hybrid computational modeling is designed to simulate a T1D progression, i.e., starting with the detection of antigens in pancreatic islets to possible outcomes of overt T1D. We focus

on modeling interactions between a class of agents including antigen, pancreatic β cell, dendritic cell, naive CD8[+] T cell, activated CD8[+] T cell, and cytotoxic CD8[+] T cell. The rules that govern interactions between these agent classes were summarized based on existing literature (which is described in detail in the **Agent behaviors** section). The selection of an agent class was based on expert knowledge and available experimental data of infiltrating CD8[+] T cells. Under different biological conditions, the model can predict the individual-specific trajectory of infiltrating CD8[+] T cells. The model can also predict the time interval between antigen detection to the possible development of overt diabetes to mimic the differentiation in T1D progression. Moreover, the model provides a computational platform to test the efficacy of different therapeutic strategies.

## Simulation software packages

The data-driven agent-based model was implemented in the NetLogo environment, which is a multi-agent programmable language and integrated modeling platform [14,28–30]. The primary user interface of NetLogo comprises two-dimensional (2D) grids, in which two types of agents are used to construct an agent-based model (as shown in **Fig 1**). A patch is an immobile agent type that comprises the background grids in the simulation space. Turtles usually referred to as mobile agents can interact with other turtles and maneuver over patches in the simulation space. Turtles can be classified into various types of agents for modeling different cell motility, interaction, migration, etc. and their associated attributes can be defined as state variables in the simulation. These state variables help differentiate the behaviors of individual agents and allow computational modelers to simulate the functions and/or actions of these agents regulated by the system. The user-friendly interface of NetLogo also allows modelers to define model parameters and observe simulation results.

## Simulation initial setting

We generated a 201 × 201 2D grid in NetLogo as the simulation interface, designed to reflect a 2D projection of the pancreas in NOD mice. The 2D projection of the pancreas comprises

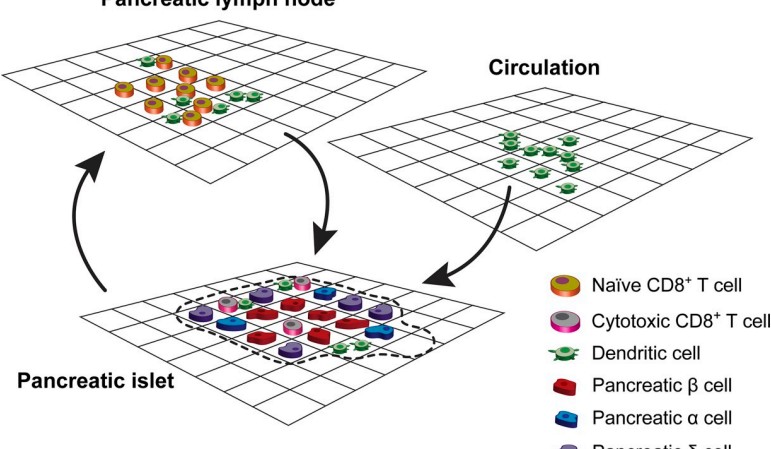

**Fig 1. Schematic illustrating pancreatic islets, circulation, and PLNs in the NOD mouse model.** Patches are shown by the background grids in NetLogo, turtles are mobile agents moving over the background grids (e.g. Naïve CD8[+] T cell). DCs and CD8[+] T cells migrate from circulation and/or PLNs to pancreatic islets, which contribute to T1D progression. One pancreatic islet is shown (the area restricted by the dashed curve in the pancreas). Pancreatic islets may overlap and form clusters of islets in the pancreas.

three areas including pancreatic islets, circulation, and PLNs, storing the three main locations of immune responses in T1D progression (as shown in **Fig 1**). A pancreatic islet is a complex zone where pancreatic β cells, pancreatic α cells, and pancreatic δ cells are located [31]. In pancreatic islets, β cells are targets of infiltrating CD8[+] T cells, and a persistent assault to β cells results in T1D [32,33]. Approximately 900,000 pancreatic β cells reside in the pancreas of the NOD mouse [26,27]. To further improve the performance of computational complexity, we simulated approximately one percent of the total number of β cells in the agent-based model. The number of other cell types was also proportionally reduced to accurately imitate the number of interacting cells. We used multiple patches (i.e. grids in the 2D NetLogo simulation interface) to represent pancreatic β cells because β cells are immobile during T1D progression.

To initialize the pancreatic β cells, 25 patches were selected as the centers of islets, and pancreatic β cells were then formed by an automated random process to furnish the area in a circular fashion with a radius of 38 grid units away from the selected center patches. The averaged diameter of a pancreatic islet is equivalent to 100 μm [27], and thus we set the length of each grid to 1.32 μm in pancreatic islets. For the simulation size presented in this paper, the number of pancreatic β cells was fixed to 8080 counts. PLNs are distributed in the surrounding area of the pancreas where APCs interact with naïve CD8[+] T cells [34–36]. We divided the entire interface of NetLogo into three regions to mimic pancreatic islets, nearby circulations, and PLNs, as illustrated in **Fig 1**. For the *in-silico* modeling environment, the random process for which different agents (cells) interact is more important than the actual physical morphology, which in *vivo* determines how these agents will interact. The choice of agents is directly comparable to the cell types and tissue organization formed in the pancreas [14]. Therefore, the NetLogo setup is appropriate for this model.

## Agent behaviors

Autoantigens, a trigger to initiate the onset of T1D, are the first agents released by apoptotic β cells [33]. Apoptotic β cells in this manuscript refer to those β cells that are in the process of programmed cell death. In accordance with the autoantigen stimuli, the number of apoptotic β cells is initialized in the agent-based platform, and then infiltrating CD8[+] T cells in the simulating experiment engage in autoimmunity. If the immune tolerance is unable to restore homeostasis for infiltrating CD8[+] T cells, beta cells eventually succumb to damage and apoptosis. Resident DCs in the neighborhood engulf the released autoantigens and become APCs [33]. These islet-resident APCs migrate to PLNs at 15–18 days of T1D onset [33] where they present autoantigens and activate naïve CD8[+] T cells [35]. The activation process has three stages. During the first stage, APCs have a random short-span interaction with naïve CD8[+] T cells; this process lasts 6–8 hours after PLNs hosting APCs [37]. Following the first stage, APCs establish prolonged and stable interactions (2–24 hours) with naïve CD8[+] T cells, leading to activation of naïve CD8[+] T cells [38]. Activated CD8[+] T cells experienced multiple rounds of differentiation with a time interval of 4–8 hours before they eventually egress PLNs after 3–5 days and approach pancreatic islets [38,39]. As naïve CD8[+] T cells become activated, APCs unbind from activated CD8[+] T cells, and then randomly advance to stimulate the next available naïve CD8[+] T cell. The lifespan of APCs ranges from 48 hours to 72 hours in PLN [40,41]. For this purpose, each APC was assigned state variables to indicate the time that they have already lived and follow a stochastic process within their lifespan range, a uniform distribution *unif*(48, 72). The value of the state variable was updated by 1 unit per simulation step, where each simulation step represents 1 hour in T1D progression. If the value of the assigned state variable for an APC agent in PLNs exceed 72 hours, the agent is then forced to die. "Die" in NetLogo leads to disappearance and removal of an agent in the simulation. During the lifespan of APCs, APCs

could activate multiple naïve CD8[+] T cells based upon the time required for triggering naïve CD8[+] T cells. Each binding activated CD8[+] T cell is then able to proliferate 4 to 7 new CD8[+] T cell every 5 days based upon experimental data presented in [42].

Activated CD8[+] T cells enter efferent lymph vessels [43] and then migrate to islets through circulation [44]. In pancreatic islets, activated CD8[+] T cells can be further stimulated by antigens released by pancreatic beta cells and become cytotoxic CD8[+] T cells [35]. Cytotoxic CD8[+] T cells move within/between islets following a random pattern, Brownian motion, and their velocities are in the range of 10–15 μm/min [44]. The speed of cytotoxic CD8[+] T cells was accordingly adjusted in the simulating experiment because the space between each islet was designed for a 2D grid simulation interface. Cytotoxic CD8[+] T cells destroy pancreatic beta cells via a direct interaction [32], and T1D progression proceeds as a result of the random movement of cytotoxic CD8[+] T cells. Pancreatic β cells undergo an apoptotic process after they are in conjunction with cytotoxic CD8[+] T cells [32,45]. As more pancreatic β cells endure the process of programmed cell death, the recruitment of DCs from circulation to islets is enhanced [46,47]. To this end, the number of recruiting DCs was correspondingly calibrated using mass-action kinetics, which will be discussed in detail in the **Mathematical Model** Section. Recruited DCs move within islets and engulf autoantigens through the phagocytosis process at sites of insulitis, and then migrate from islets to PLNs. Within PLNs, following these cycles of migration, the recruited APCs further activate naïve CD8[+] T cells and promote the prognosis in pancreatic T1D.

Pancreatic β cells may proliferate their offspring during T1D progression [48–50]. A recent study also found out that regenerating and glucose-stimulated β cells can re-enter cell division cycles in a shorter period, compared to β cells recovering through cell division cycles under normal conditions [51]. To mimic these characteristics of β cells in NetLogo, each β cell is assigned to a state variable with the purpose to identify a quiescence period. During the quiescence period, β cells are not able to divide and proliferate. Based upon the study [51], pancreatic β cells can remain in a quiescence period for approximately 7 days (168 hours). At the beginning of a simulation, we assume the quiescence period of β cells follows a uniform distribution (*unif*(0, 168)). If the state variable (associated with the quiescence period) of a specific β cell is equal to 72, it means that this β cell has stayed in a quiescence period for 72 simulation steps (note 1 simulation step represents 1 hour in T1D progression). During the simulation, the quiescence period of each β cell was checked every simulation step. Once the quiescence period of β cells exceeds the maximum quiescence period (e.g. 168 hours), β cells can re-enter the cell division and start to proliferate new β cells at the next simulation step. The quiescence period of β cells was shortened as the simulation proceeds since regenerating and glucose-stimulated β cells re-enter the cell division cycles in a shorter period during a T1D progression. A change in the quiescence period of β cells was formulated using a power function, as demonstrated in the **Mathematical Model** Section. **Fig 2** shows the behavior of different types of agents within islets, circulation, and PLNs during T1D progression. **Fig 3** demonstrates all the interactions between agents and the level of intelligence in the ABM. For a detailed description of agents' setup and cellular interactions in NetLogo, one can refer to **S1**, **S2** and **S3 Tables**.

## Mathematical model

To calibrate quantitative changes in agent number during T1D, we used mathematical expressions and mathematical models to measure the recruiting process of circulating DCs and the pancreatic β cell replication. For example, we calibrated the mass-action kinetics to measure the number of recruiting DCs from circulation, which depends upon apoptotic β cells,

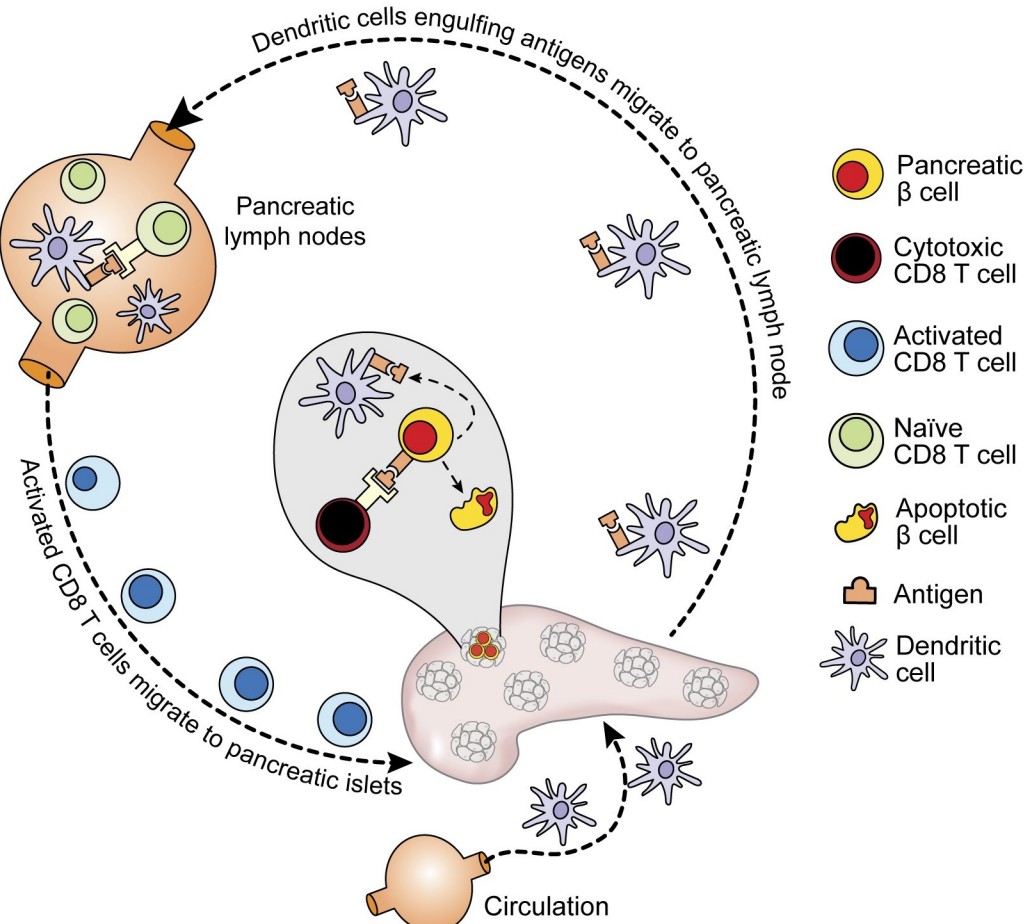

**Fig 2. Schematic diagram of T1D progression in circulation, islets, and PLN.** The legend shows the cell types and autoantigens used as agents in the ABM structure.

expressed as follows:

$$DC_r|_{t=j+1} - DC_r|_{t=j} = k_r \beta_{AP}|_{t=j} DC_c$$

where $DC_r|_{t=j+1}$ represents the number of recruiting DCs at the simulation step $j+1$, and $DC_r|_{t=j}$ represents the number of recruiting DCs at the simulation step $j$. $\beta_{AP}|_{t=j}$ represents counts of apoptotic β cells at the simulation step $j$, $k_r \in N(\mu_r, \sigma_r)$ represents the recruiting rate for DCs in circulation for a given $(\mu_r, \sigma_r)$, and $DC_c$ is defined as counts of DCs in circulation. The number of circulating DCs during T1D progression was observed to remain unchanged [52]. $DC_c$ was then assumed to stay stable in the simulation.

Experimental studies reported that β cells had an increased rate of proliferation, functional recovery, and resistance to autoimmune destruction during T1D progression in a NOD mouse model [48,50,53,54]. A recent study also suggested that replicated β cells were able to re-enter cell division after a certain quiescence period [51]. This study also found out the quiescence period was shortened by an increased rate of glucose metabolism [51]. In addition, recent studies suggested that the glucose level was enhanced as the percentage of apoptotic β cells increased [55]. Based upon the experimental evidence, we assume that the quiescence period of viable β cells (the apoptotic β cells and dead β cells lose the ability to replicate) is associated

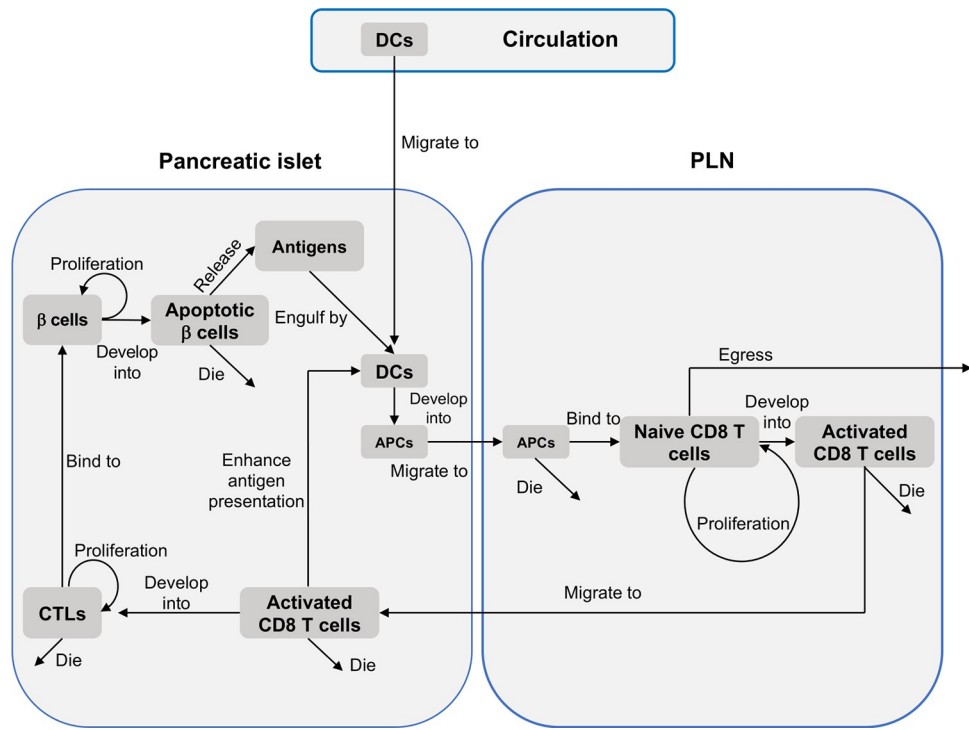

**Fig 3. Components of the agent-based model.** The diagram demonstrates all the interactions between agents in circulation, islets, and PLN and illustrates the level of intelligence in the ABM.

with the percentage of apoptotic beta cells in a time-dependent manner. To model β cell replication, functional recovery, and resistance to autoimmune destruction, we propose the following mathematical expression:

$$T_{\beta_i}\big|_{t=j} = \left(1 - \frac{\beta_{AP}\big|_{t=j}}{\beta_{init}}\right)^{\gamma} \times T_{\beta_i}\big|_{t=0}$$

where $T_{\beta_i}\big|_{t=j}$ represents a quiescence period (the quiescence period represents a combined effect of replication of β cells, functional recovery of β cells, and β cells' resistance to autoimmune destruction) of the $i_{th}$ β cell at the simulation step $j$, and the power $\gamma$ describes the degree of glucose metabolism in the system. The degree of glucose metabolism was estimated by counts of apoptotic β cells, $\beta_{AP}$, the higher the apoptotic β cells in the system, the shorter the quiescence period of surviving β cells. $\beta_{init}$ represents counts of healthy β cells at the onset of T1D autoimmunity, simulation step = 0. The value of $\gamma$ was estimated and set to 1 because when the quiescence period of β cells becomes 2 days there are 70% of apoptotic β cells in the islets ($0.3^{\gamma} \times 7 = 2$)) [51]. $T_{\beta_i}\big|_{t=0}$ represents the quiescence period of β cells under normal condition (simulation step = 0). A state variable that calculated how long each β cell survived in islets was assigned to each β cell at the beginning of the simulation. A β cell started to undergo the replication process if the value of this state variable exceeded the value of $T_{\beta_i}\big|_{t=j}$.

## Experimental data collection

In addition to mathematical models, we collected experimental data such as the percentage of cytotoxic CD8[+] T cells proliferating within islets, the time required for DCs migrating from islets to PLNs, and the lifespan of naïve CD8[+] T cells in PLNs from existing experimental studies. These data were incorporated into data-driven agent-based modeling as system parameters. Most of the data we collected were time-related experimental data, each of which helped to simulate kinetic analysis of the model. We collected experimental data from studies that were most similar to our simulation setting (*e.g.* in a NOD mouse model). We also investigated other parameters used in simulations if data were not available from experimental studies, which is explicitly described in the next section. A summary of the collected experimental data is provided in **Table 1**.

## Parameter estimation

The Latin hypercube sampling (LHS) method is applied to estimate the default values of unknown parameters. The ranges of unknown parameters were summarized from literature and obtained from the field experts' suggestions, which are included in **S4 Table** (i.e. Ranges used for LHS). Based on the LHS method, the range of each parameter value is divided into $n = 100$ intervals and each interval of a parameter is sampled once. A total of 100 combinations of parameters generated from the LHS Package in R was performed on our local server to obtain the dynamics of CD8[+] T cells and compare them with the experimental data. About 25 simulations were run simultaneously each time on the server (server-specific configuration: 4x Intel Xeon CPU E5-4650 2.1 GHz 48 cores with 192 GB of RAM). In addition, each set of parameters was then performed 20 times to obtain the average counts of infiltrating CD8[+] T cells at week 4, week 6, week 8, week 10, week 12, and week 14. A set of parameter values was selected as the default values that satisfy the condition where the difference between the simulated results and experimental data at weeks 4, 6, 8, 10, 12, and 14 is optimally minimized. The

**Table 1. Data from experimental studies in the NOD mouse model.**

| Experimental data | Value | Sources |
|---|---|---|
| Percentage of CTLs proliferating in islets | (15.4–23.8) % | [42] |
| Number of islets in the pancreas | 2500 counts | [61] |
| Lifespan of naïve CD8[+] T cell in PLN | (3–5) days | [38,39] |
| Time required for DCs migrating from islets to PLNs | (15–18) days | [33] |
| Lifespan of DCs in PLN | (48–72) hrs | [40,41] |
| Time required for naïve CD8[+] T cell activation in PLN | (2–24) hrs | [37] |
| Time required for β cells remaining in a quiescence period under a non-glucose-simulated environment | 168 hrs | [51] |
| Time required for β cells remaining in a quiescence period under a glucose-simulated environment | 48 hrs | [51] |
| Time required for DCs interacting with naïve CD8[+] T cells | (6–8) hrs | [37] |
| Time required for naïve CD8[+] T cell differentiation in one cycle | (4–8) hrs | [38,39] |
| Time required for β cell replication | 24 hrs | [51] |
| Time required for CTL differentiation | 120 hrs | [42] |
| Number of CTL proliferating | (4–7) counts | [42] |
| Islet diameter | (100–110) μm | [27] |
| Time required for activated CD8[+] T cell migrating from PLN to islets | 120 hrs | [62] |

results then showed that kinetics of CD8$^+$ T cells highly follow oscillatory dynamics as observed and suggested by studies and experts [56–58]. Parameter ranges utilized for local sensitivity analysis are included in **S4 Table**. The ranges determined by the field experts are marked using asterisks. For other ranges, appropriate references are provided in **S4 Table**.

### Local sensitivity analysis

An imperative characteristic of modeling studies of physiological systems, such as T1D, is the potential to discover how sensitive (or uncertain) different mechanisms of the model are to variabilities and thus pinpoint promising targets for therapeutic purposes. One aspect of the sensitivity analysis is used to determine how elevated the change in a given output is generated by perturbations in model input values [59,60]. The sensitivity analysis provides modelers insights into the inherent nature of the model by investigating the relationship between input and output variables. Within the sensitivity analysis, local sensitivity analysis quantifies the effect of small variations of input factors on output variables when only one factor is changed at a time [59]. A global sensitivity analysis was conducted later to investigate the effect of multiple interactions between the inputs and model output. Compared to the global sensitivity analysis (as described in the next section), local sensitivity analysis requires less computational costs [60].

For our simulating experiment, we deal with a total of $N_P = 21$ unknown parameters in the model (as tabulated in **S4 Table**). Local sensitivity analysis provides initial screening for sensitive parameters, and then helps target the most sensitive parameters for global sensitivity analysis. To conduct a local sensitivity analysis, one factor was altered at a time while other factors remained unchanged. For the varying parameter, nine values, $M = 9$, were selected on either side of the default (baseline) value, $P_d$. Default values were estimated to fit the experimental data (details were presented in the **Result** section) by numerous simulation replications. Four values were selected on the left side of the default value, and four values on the right side of the default value. The interval between the selected values was set equally to 5% of the default value (in total the values range between $[P_d \pm 20\% \times P_d]$). Each selected value, $P_s \in [P_d \pm 20\% \times P_d]$, was assumed to follow a normal distribution. The mean of the normal distribution was consecutively set to each of the nine selected values, and the standard deviation was calculated as 6% of the default value of the selected parameter (the maximum value of each parameter can be found in **S4 Table**), to ensure the parameter values extend to the larger range $[P_d \pm 30\% \times P_d]$. Each selected value was performed $N = 20$ simulation runs to capture the stochastic nature of the agent-based model. For these simulation runs, the time required for an overt T1D was recorded as an output variable of interest $X_s = X_s(P_s)$.

For each selected value of tested parameters, a boxplot was generated to measure 25th percentiles, median, and 75 percentiles of the output variable, as formulated below:

$$X_{s_j} = [X_{s_j}^1, X_{s_j}^2, \dots, X_{s_j}^N] \text{ for } j = 1, 2, \dots, M$$

$$Q_1\left(X_{s_j}\right) = X_{s_j}\left(\left\lceil \frac{25}{100}N \right\rceil\right)$$

$$Q_2\left(X_{s_j}\right) = \begin{cases} \dfrac{X_{s_j}\left(\dfrac{N}{2}\right) + X_{s_j}\left(\dfrac{N}{2}+1\right)}{2}, \text{ if } N \text{ is odd} \\ X_{s_j}\left(\dfrac{N+1}{2}\right), \text{ if } N \text{ is even} \end{cases}$$

$$Q_3\left(X_{s_j}\right) = X_{s_j}\left(\left\lceil \frac{75}{100}N \right\rceil\right)$$

where $X_{s_j}$ is a sorted vector, that is $X_{s_j}^{i+1} \geq X_{s_j}^{i}, \forall i = 1, 2, \cdots, N-1$, representing a series of data, where $X_{s_j}^{i}$ is the $i_{th}$ data in the sorted vector. $Q_1$, $Q_2$, and $Q_3$ denotes the 25$^{th}$ percentile, the median, and 75$^{th}$ percentile of the sorted vector, respectively. By calculating the quartiles of a sorted vector, the distribution of data can be easily evaluated.

Furthermore, a one-way ANOVA test [63] was performed to identify if different levels of unknown parameters have different effects on the model output (e.g. time required for T1D development). The ANOVA test helped identify the sensitivity of unknown parameters to the model output, which was conducted in the following procedures:

$$H_0: \ \mu_1 = \mu_2 = \cdots = \mu_M \ \text{and} \ H_a: \ \text{not all mean values are equal}$$

where $\mu_j, j = 1,2,\ldots,M$ represent the means of the output variable based upon each selected value $P_s$ of unknown parameters and the F-values is obtained from [63]

$$F = \frac{SSR}{SSE} = \frac{\sum_k n_k(\bar{y}_{\cdot k} - \bar{y}_{\cdot \cdot})^2 n_k(\bar{y}_{\cdot k} - \bar{y}_{\cdot \cdot})^2/(M-1)}{\sum_i \sum_k (y_{ik} - \bar{y}_{\cdot k})/(N-M)} \sim F_{M-1, N-M}$$

where $y_{ik}$ represents the $i^{th}$ observation in $k^{th}$ group. In ANOVA tests, the total variation in the data is partitioned into two components: $SSR$ and $SSE$, representing the sum of squares due to the between-groups effect, and the sum of squared errors, respectively. $(y_{ik} - \bar{y}_{\cdot k})$ is a variation of observations in each group from their group mean estimates, $\bar{y}_{\cdot k}$, (i.e. variation within the group), $(\bar{y}_{\cdot k} - \bar{y}_{\cdot \cdot})$ denotes variation of group means from the overall mean, $\bar{y}_{\cdot \cdot}$, (i.e. variation between groups), and $n_k$ is the sample size for the $kth$ group. One should reject the null hypothesis $H_0$ if the p-value of the $F$ test is smaller than the predetermined significance level (e.g. $\alpha = 0.05$), which indicates that at least one of the mean values of the output variable (i.e. time required for T1D development) is significantly different from others.

## Global sensitivity analysis

Global sensitivity analysis is a technique applied to simulations of the performance and the quantitative effect of input variables on output variables when the input variables are varied over the entire possible ranges [60]. Both the extended Fourier amplitude sensitivity test (eFast) and Sobol's method deliver the measures for the decomposition of the total variance of model output into the main and interaction effect of each parameter. Compared to Sobol's method, eFast is more computationally efficient because it requires fewer sampling/simulations [60]. As such, we implemented an eFast, which is an advanced version of the Fourier amplitude sensitivity test (FAST). As introduced in the 70s, FAST is proposed to implement sensitivity analysis for both monotonic and nonmonotonic models. Compared to other sensitivity analyses such as partial correlation coefficients, FAST can more efficiently explore the multidimensional space of input factors by a suitably defined search-curve [64]. The search curve for the input factor $P_i$ is defined as follows:

$$q_i = \frac{1}{2} + \frac{1}{\pi} arcsin(sin \ \omega_i s) \ \text{for} \ i = 1, 2, \ldots, N_l \tag{1}$$

where $q_i$ representing a search curve in the unit interval ($q_i \in [0,1]$ for $i = 1,2,\ldots,N_l$) was proposed by Saltelli *et al.* [64], and $N_l = 5$ representing the number of sensitive parameters based

on local sensitivity analysis. The search-curve generates normalized sample points more uniformly distributed in the unit interval [64]. In Eq (1), $\omega_i$ for $i = 1,2,\ldots,N_l \leq N_P = 21$ denotes the corresponding frequency for $q_i$ corresponding to the normalized value of $i^{th}$ input factor and $s$ varies within the interval $(0, 2\pi)$. The selection of $\omega_i$ starts from the determination of $\omega_{max}$ for a series of input factors such as $Q = \{q_i, i = 1,\cdots,N_l\}$. A maximum frequency $\omega_{max}$ for the series of input factors can be obtained by the equation $\omega_{max} = \frac{n-1}{2m}$, where $n$ is the predetermined sample size (e.g. $n = 200$ for our simulation study), $m$ denotes the interference factor (a default value is 4) [65]. For the remaining frequencies, the maximum allowable frequency is given by: $\omega'_{max} = \frac{\omega_{max}}{2m}$, where $\omega'_{max}$ represents the second maximum frequency for the factor set $Q$. This strategy for selecting other frequencies is applied to ensure the search curves has distinct frequencies for $\omega_{max}$ and $\omega'_{max}$. Thus, the search curve can fill the sampling space as much as possible. For a detailed description, one can refer to an automated algorithm proposed by Saltelli *et al.* [64]. Moreover, since the range of the input factor $q_i$ is from 0 to 1, we implemented a quantile function to transform the unit factor $q_i$ to a realistic value of input factors $X_i$. **S2 Fig** illustrates the 2D transformation from the search curve to the sample points using eFast.

In **S2A Fig**, two search curves were depicted, with a search frequency equal to $\omega_1 = 24$ and $\omega_2 = 1$, respectively. Then a scatterplot of sampling points for two factors is shown in **S2B Fig**. Each point in the space needs to be transformed to a realistic value using a quantile function, as shown in **S2C Fig** in which a cumulative distribution function (CDF) of a normal distribution with the random variable $X \sim N(100, 20)$.

eFast is developed based upon the original FAST, and the advantage of eFast over the original FAST is the ability to evaluate not only the first-order sensitivity index (i.e. effect of one factor) but also the total-order sensitivity index (i.e. the interaction effect between input factors), and how these interaction effects may affect the model outputs [60,66]. The first-order sensitivity index and the total-order sensitivity index for $i = 1,\cdots,N_l$ can be obtained using the following equations:

$$A_j = \frac{1}{2\pi} \int_{-\pi}^{\pi} f(s)\cos(js)ds \tag{2}$$

$$B_j = \frac{1}{2\pi} \int_{-\pi}^{\pi} f(s)\sin(js)ds \tag{3}$$

$$\Lambda_j = A_j^2 + B_j^2 \tag{4}$$

$$\hat{D}_i = 2\sum_{p=1}^{m} \Lambda_{p\omega_i} \tag{5}$$

$$\hat{D}_T = \hat{D} - 2\sum_{j=1}^{\omega_i/2} \Lambda_j \tag{6}$$

$$\hat{D} = 2\sum_{j=1}^{(n-1)/2} \Lambda_j \tag{7}$$

$$S_i = \frac{\hat{D}_i}{\hat{D}} \text{ and } S_T = \frac{\hat{D}_T}{\hat{D}} \tag{8}$$

Eqs (2)–(8) define the Fourier coefficients $A_j$ and $B_j$ for $i = 1,\cdots,N_l$ in a Fourier series, where the real-valued function $f(s)$ represents the simulation results that were expanded by a Fourier series [64]. The reason why the Fourier coefficients were introduced is that

partitioning of variance in eFast works by varying different parameters at different frequencies [66]. $\hat{D}_i$ for $i = 1, \cdots, N_l$ represents a fraction of total variances induced by the uncertainty of $i_{th}$ input factor. $\hat{D}_T$ calculates the variances induced by the input factor and the interactions between the input factor and other input factors (first + higher-order). $\hat{D}_T$ can be calculated by the difference between the total variance $\hat{D}$ and the complement term $2\sum_{j=1}^{\omega_i/2} \Lambda_j$, as given in Eq (8). The complement term helps calculate the variance produced by other frequencies than $\omega_i$. The ratio $\frac{\hat{D}_i}{\hat{D}}$ was applied to estimate the main effect of $i_{th}$ input factor on the output, and the ratio $\frac{\hat{D}_T}{\hat{D}}$ was employed to calculate the main and interaction effects of $i_{th}$ input factor on the output, respectively.

### Therapy-directed simulations

Therapy-directed simulations were conducted to verify the effects of hypothetical therapies on the output variable, the time required for T1D development. The goal of implementing therapy-directed simulations is to identify possible therapeutic strategies that could reduce the chance of T1D development. For the simulation purpose, T1D was developed if the survival percentage of healthy β cells falls within the range of 10% - 30% of the initial healthy β cells (the total number of β cells at the simulation step 0) [45,48]. The hypothetical therapies are proposed based upon the results from both local and global sensitivity analyses. For the therapy-directed simulations, 50 simulation runs were conducted for each strategy, and the probability of developing T1D was calculated using the following formulation:

$$P_{\text{T1D}} = \frac{\textbf{Number of T1D cases observed in simulations}}{\textbf{Total number of simlations}}$$

where $P_{\text{T1D}}$ represents the probability of leading to T1D under the proposed strategy.

## Results

### Validation of model using experimental study

To cross-validate the agent-based model, we compared the number of infiltrating CD8+ T cells from multiple simulation runs to the number of infiltrating CD8+ T cells collected in the experimental study [62]. Magnuson et al. [62] studied the population dynamics of infiltrating CD8+ T cells in a NOD mouse model. They measured the number of infiltrating CD8+ T cells at week 4, week 6, week 8, week 10, week 12, and week 14, respectively, and provided us the raw data for model validation (please refer to **Acknowledgment Section**). Our simulated results demonstrated that the number of infiltrating CD8+ T cells have an agreement with the experimental data in a time-dependent manner ($R^2$ = 0.98 calibrated by comparing the mean values between model fit and experimental data), as shown in **Fig 4**.

As illustrated in **Fig 4**, a significant increase in infiltrating CD8+ T cells occurs around week 8 in the simulation, which is consistent with the findings of the previous studies [67,68]. During the time course of T1D progression, activated CD8+ T cells continue to migrate from pancreatic lymph nodes (PLNs) to the site of islets, which causes persistent damage to β cells and eventually leads to overt T1D after week 12 (see **S1 Movie**). The oscillatory dynamics and kinetics of CD8+ T cells have been postulated to follow by a series of immunological responses including the death of CD8+ T cells in islets, the proliferation of CD8+ T cells within islets, and the recruitment of CD8+ T cells from PLNs to islets due to beta-cell proliferation (see the agent rules presented in **S2 Table**), which was also observed and proposed by Trudeau *et al.*, Vendrame *et al.*, and von Herrath *et al.* [56–58]. These immunological responses were

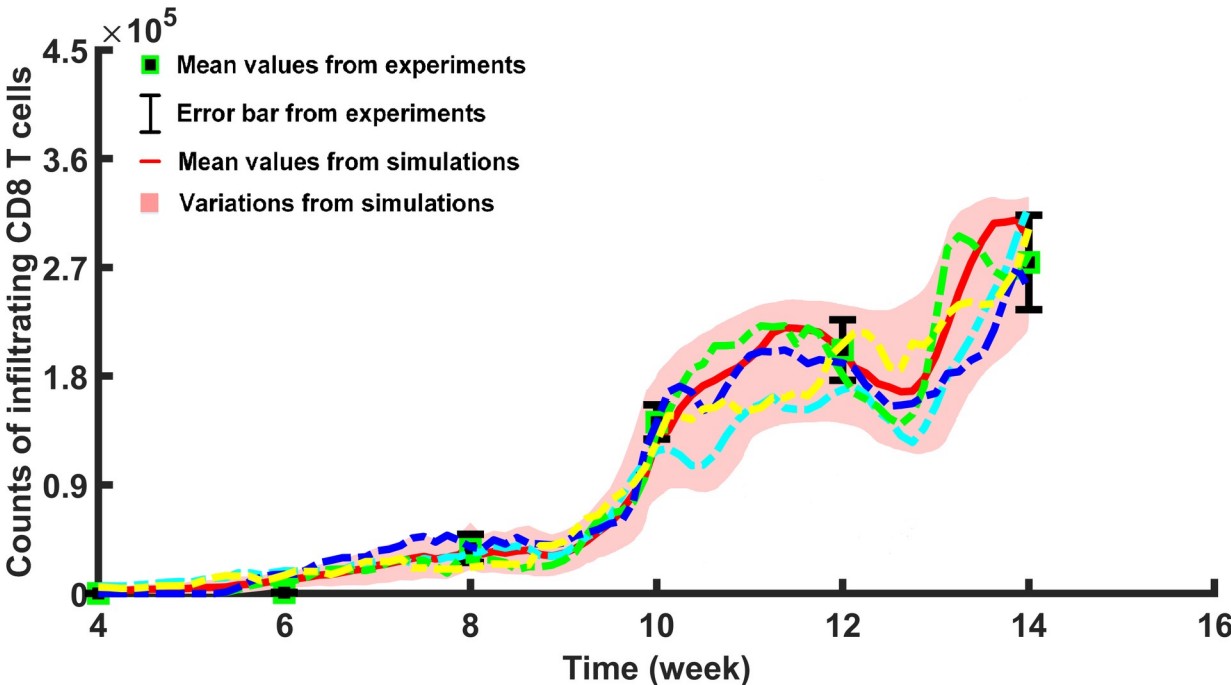

**Fig 4. Individual-specific kinetics of infiltrating CD8+ T cells in T1D progression.** This figure shows a comparison of infiltrating CD8+ T cells between one hundred simulation runs and the experiment at week 4, week 6, week 8, week 10, week 12, and week 14, respectively. In this experiment, average counts of infiltrating CD8+ T cells (denoted by the red line) were calculated based upon 100 simulation runs (mean ± SEM and the shaded area represents variations in the simulation runs). Four representative trajectories of infiltrating CD8+ T cells (shown by dashed lines in blue, green, yellow, and cyan) reflect individual-specific patterns in the inherently noise-driven T1D progression. In addition, infiltrating CD8+ T cell counts (shown by the green square for mean values and black error bars for variations in the experiments) were illustrated using the raw data (7 replications at week 4 and week 12, 6 replications at week 6 and week 10, 5 replications at week 8, and 12 replications at week 14).

incorporated in the ABM framework and average counts of infiltrating CD8+ T cells (denoted by the red line and illustrating cyclic behavior) were calculated based upon 100 simulation runs (mean ± SEM). Furthermore, regulated patterns with individual variations in model results concur with experimental data reflecting cell fate and population heterogeneity demonstrated that T1D progression implies an inherently noise-driven process but a highly orchestrated and robust physiological mechanism. Besides, considering 21 unknown parameters in the system, the number of possible combinations is extremely large and, therefore, a Latin Hypercube Sampling (LHS) [69] was applied to efficiently scan parameter spaces and help estimate accurate values and distributions of unknown parameters. By minimizing the difference between the simulated results and experimental data ($R^2$), the default values of unknown parameters were estimated and listed in **S4 Table**.

To understand the effect of infiltrating CD8+ T cells on pancreatic β cells, time-dependent counts of pancreatic β cells were replicated. As depicted in **Fig 5A**, counts of healthy β cells start to significantly decrease at around week 8 (when the simulation in ABM interface shows 1344 steps equivalent to 1344 hours; note that 1 simulation step represents 1 hour in T1D progression). The survival analysis of β cells in pancreatic islets persistently decreases, accompanying a continuous increase of infiltrating CD8+ T cells in islets (as shown in **Fig 4**). In the meanwhile, discernible increases in β cell counts occurring after week 12 were also detected following repeated decreases. This phenomenon is referred to as the "Honeymoon Period" that may be induced by β cell regeneration, resistance to autoimmune destruction, and functional recovery during T1D progression [9,10,12,49,50,53,54] in congruence with experimental studies which was observed in [56,70–72].

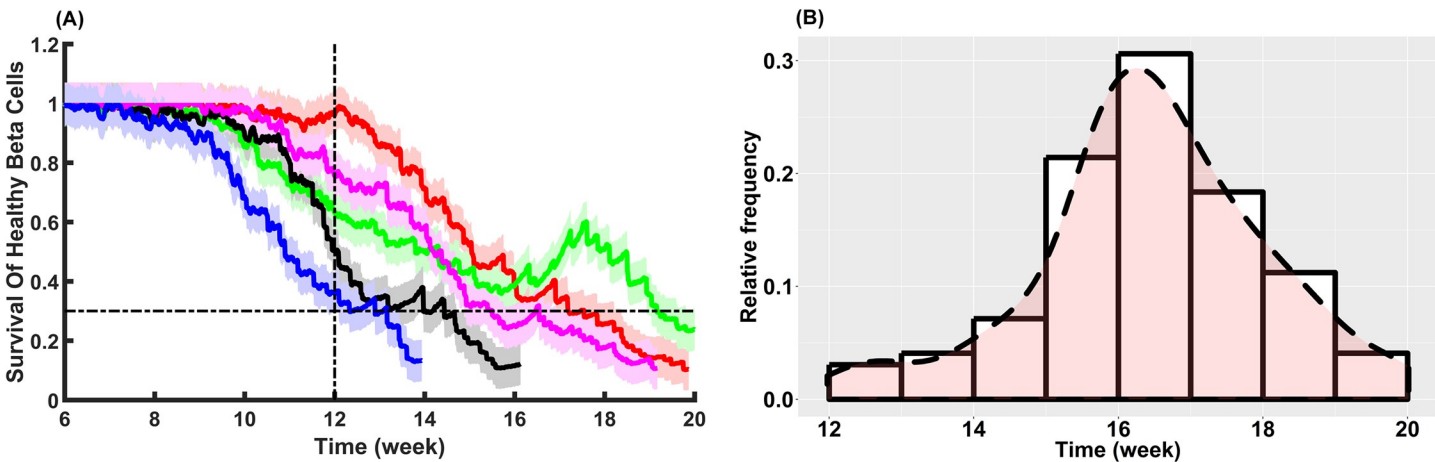

**Fig 5. Validation of the ABM model using experimental findings. (A)** Survival of healthy β cells during individual-specific T1D progression. Five solid lines represent the enduring fraction of healthy β cells in 100 distinct simulation runs with the same initial setting ($n_i$ = 20 and mean$_i$ ± SEM$_i$, for $i \in$ {1: Blue, 2: Black, 3: Green, 4: Magenta, 5: Red}; parameter values can be found in **S1** and **S5 Tables**). The vertical dotted-dashed line represents week 12 after the simulation initialization. The horizontal dotted-dashed line ($y$ = 0.3) denotes the remaining 30% of healthy β cells, about which overt T1D may occur. **(B)** Distribution of overt T1D development occurring during week 12 and week 20 in 100 simulation runs corroborating experimental data [45,73,74]. The horizontal axis represents the time required for developing overt T1D in-silico, and the vertical axis denotes the relative frequency of T1D incidence. It is important to note that T1D onset occurs in our simulations based on the assumption that the survival percentage of healthy β cells falls randomly within the range of 10% - 30% of the initial healthy β cells (the total number of β cells at the simulation step 0).

Although there were no experimental data for the dynamics of β cells in the NOD mouse model from the same study of T-cell dynamics to directly verify the results in **Fig 5**, the model shows that it captures the onset of overt diabetes in good agreement with the distribution of the time at which NOD mice become diabetic extracted from the existing literature [45,48,57,73–75]. The time at which NOD mice become diabetic is heterogeneous, and most diabetes incidences in NOD mouse models occur during week 12 and week 20 [45,73,74]. Simulated results corroborating the highly regulated heterogeneity in T1D incidences demonstrate, as shown in **Fig 5B**, that T1D development takes place most frequently between week 12 and week 20 after the onset of autoimmunity. It is also worthwhile to point out that, based upon 100 simulation runs, 50% (50 out of 100) of T1D cases occur during week 16 and week 18. Analogous to the NOD mouse model, onset of disease in the ABM model is not an age-dependent event, and the time at which T1D becomes overt in the simulations is assumed that the survival percentage of healthy β cells falls randomly within the range of 10% - 30% of the initial healthy β cells (the total number of β cells at the simulation step 0).

## Sensitivity analysis

One of the major challenges of modeling a complex biological system is the lack of sufficient experimental data. Constructing a standardized and reliable computational procedure from statistical learning to analyze the current biological knowledge and available experimental data is therefore essential. For this purpose, we propose to investigate such a lack of information using ABM to quantitatively simulate the performance of phase progressions in complex pancreatic cellular networks during T1D autoimmunity. In addition, numerous essential factors and parameters are unknown (as demonstrated in **S4 Table**) during T1D progression. By implementing the sensitivity analysis, the effects of variations and heterogeneities of unknown parameters on the output of interest (e.g. the time required for overt T1D occurrence) can be thoroughly studied.

We initially investigated the impacts of unknown parameters on the outcome of T1D progression by implementing local sensitivity analysis to identify the most sensitive parameters that perturb significantly dynamics of the complex system. As a primary screening method, local sensitivity analysis provides information on how heterogeneity in each factor changes the profile and behavior of T1D autoimmunity. Note that the local sensitivity analysis implemented here is employed by fluctuating the magnitude of one parameter but keeping values of other parameters fixed [76]. The underlying assumption of the local sensitivity analysis is that the relationship between input parameters and output profiles of the model is undeviating and almost linear when the change in input parameters is relatively small. In ABM, local sensitivity analysis is often achieved by varying selected inputs within their confidence intervals based upon a specific percentage surrounding their default values [60]. To accurately illustrate the effects of unknown parameters on output profiles of the model, we conducted 20 simulation replications for each selected parameter value for the local sensitivity analysis (details on value selection were described in Section **Local sensitivity analysis**). A summary of results from the local sensitivity analysis is illustrated in **Fig 6**. Among all the unknown parameters, five sensitive parameters are presented in **Fig 6** and the remaining boxplots for the spectrum of insensitive parameter values can be found in **S4 Fig**.

Using the local sensitivity analysis, boxplot-and-raw-data distributions of T1D occurrence were captured and plotted in **Fig 6**. By means of a one-way analysis of variance (ANOVA) test, we identified that 1) the average lifespan of cytotoxic CD8[+] T cells (CTLs) within the pancreatic islets shown in **Fig 6A**; 2) the initial number of damaged beta cells when the simulation step equals 0 shown in **Fig 6B**; 3) time interval of dendritic cells (DCs) movement in islets shown in **Fig 6C**; 4) the maximum number of naïve CD8[+] T cells binding to DCs shown in **Fig 6D**; 5) recruitment rate of DCs within the pancreas shown in **Fig 6E** are the five sensitive parameters during T1D progression.

We only focused on T1D incidence within 20 weeks after the onset of autoimmunity which was suggested by experimental studies [45,73,74], and if no overt T1D was observed within this time interval, we reported the highest possible range for simulation run which is set to 32 weeks. As depicted in **Fig 6**, we discovered that a delay in the T1D occurrence is virtually inevitable, revitalizing survival of healthy β cells if the following conditions meet: 1) recruitment rate of DCs reduces; 2) the maximum number of naïve CD8[+] T cells binding to DCs reduces; 3) time interval of DCs movement increases. Moreover, we observed that T1D progression may be inhibited if the average lifespan of cytotoxic CD8[+] T cells in islets is reduced to five days or less, or the number of apoptotic β cells is less than a certain threshold estimated at most 510 damaged beta cells exist when the simulation begins (the range for the initial number of damaged cells has been scaled up to reflect the actual size of β cells). It is also worthwhile to point out that acute and progressive onset of diabetes chronicled by different studies through differentiation of T1D phenotype in the NOD mouse model [10,77,78] were also captured in our sensitivity analysis as shown in **Figs 6** and **S4**.

The local sensitivity analysis provides interesting modeling insights by changing one parameter at a time. However, the output profiles of the simulation can be significantly changed by systematic variations in key model parameters. Thus, it is worthwhile to investigate how variations in the input parameters change the model output due to their interaction with other parameters. To further decipher the interaction effects between parameters, we applied the variance-based extended Fourier amplitude sensitivity test (eFAST) method for global sensitivity analysis. To this end, five sensitive parameters were selected based upon the results from local sensitivity analysis to perform eFAST. After transforming search curves into physiological ranges (as shown in **S2 Fig**), sampling points of the five sensitive parameters based upon various frequencies are drawn from their distributions as illustrated in **S3 Fig**.

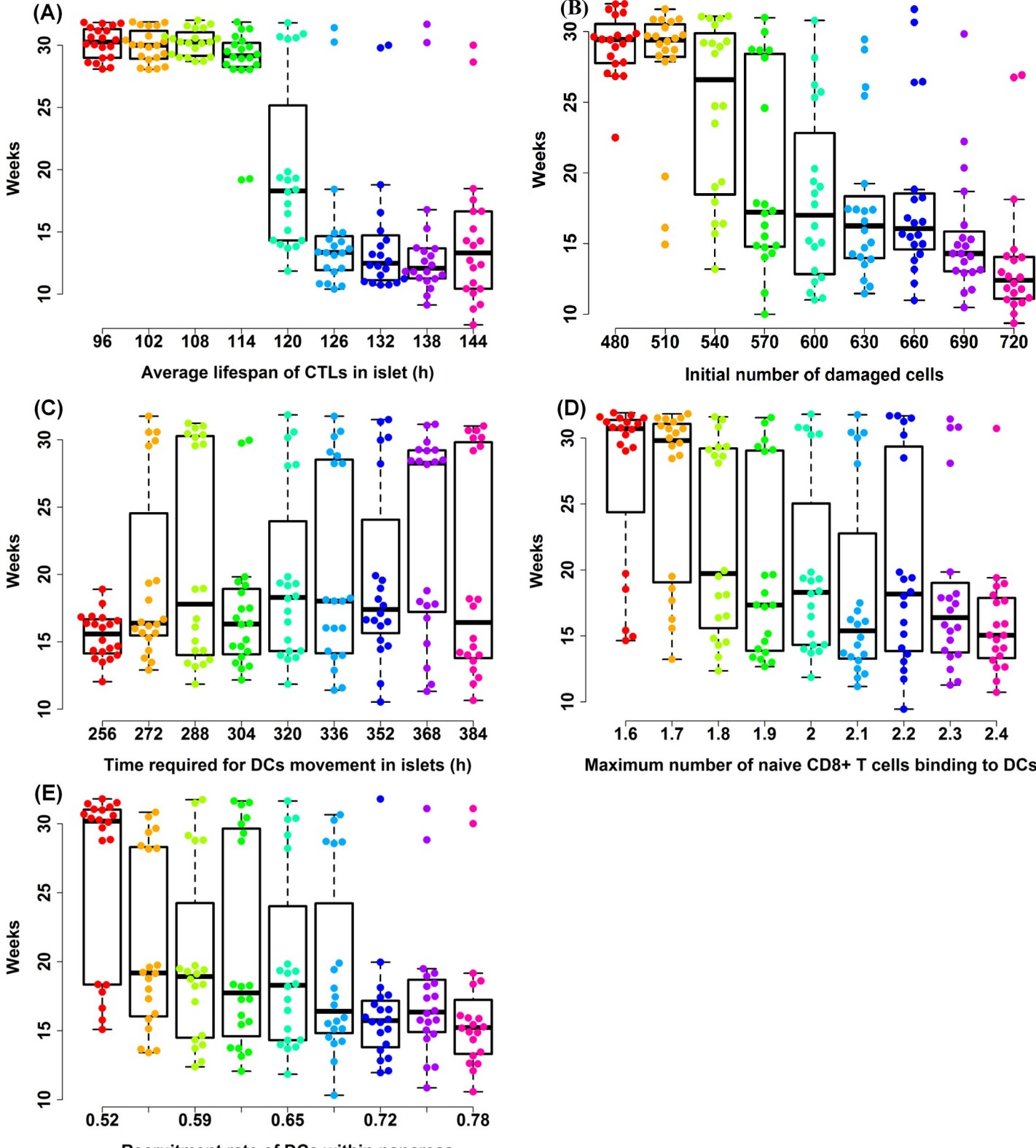

**Fig 6. Discernible fluctuation in time required for developing overt T1D based upon selected values of parameters using local sensitivity analysis. (A-E)** The horizontal axes represent selected values of five input parameters in the spectrum of $[P_d \pm 30\% \times P_d]$, and the vertical axes denote the time required for developing T1D. Box plots and raw data illustrate heterogeneities occurring during T1D for selected values. The black dashed lines outline the lower and upper whiskers and black solid

boxes show $Q_1$, $Q_2$, $Q_3$ for the first quantile, interquartile, and third quantile values for selected points within the spectrum of $[P_d \pm 30\% \times P_d]$. For these five sensitive parameters (shown by labels of the horizontal axes), the p-values of F tests were smaller than the predetermined significance level (e.g. $\alpha = 0.05$).

Using **Eqs (2)**–(**8**), the first-order index and total-order index were calculated, and the main effects and interaction effects of the five selected parameters on the model output are shown in **Fig 7**. The model output in these simulations, as aforementioned, constitutes the time required for developing overt T1D in NOD mice. If T1D was not detected within 20 weeks of autoimmunity (i.e. the number of remaining healthy β cells exceeds a threshold within the range of 10%-30% of initial β cell counts at each simulation), we assume no T1D occurrence was observed for this experiment concurring with experimental studies [45,73,74].

In this case, as shown in **Fig 7A**, the full data set comprised all possible outcomes of the model output in each simulation run, whether T1D occurs within 20 weeks of autoimmunity or not. As a result, the lifespan of cytotoxic $CD8^+$ T cells plays a key role in the global sensitivity analysis of the model output since the first-order index of this parameter approaches 70% (sensitivity of parameter $P_1$ shown in **Fig 7A**). Corroboration for this hypothesis and conclusion stems from the fact that the elongated lifespan of cytotoxic $CD8^+$ T cells in islets is directly correlated to an increase in the number of cytotoxic $CD8^+$ T cells during T1D progression. Less cytotoxic $CD8^+$ T cells survive during T1D disease if the average lifespan of cytotoxic $CD8^+$ T cells is reduced over time. T1D progression is significantly delayed or even inhibited, which may lead to a promising therapeutic target if a limited number of cytotoxic $CD8^+$ T cells circulate in the islets (also observed in **Fig 6A**). In contrast to the average lifespan of cytotoxic $CD8^+$ T cells in islets, other input factors such as the maximum number of naïve $CD8^+$ T cells binding to DCs and the recruitment rate of DCs circulating in the pancreas experience minimal effects on sensitivity analysis of the model output for first-order indexes, while they exhibit approximately 25% contribution to the model output when the interaction effects were included.

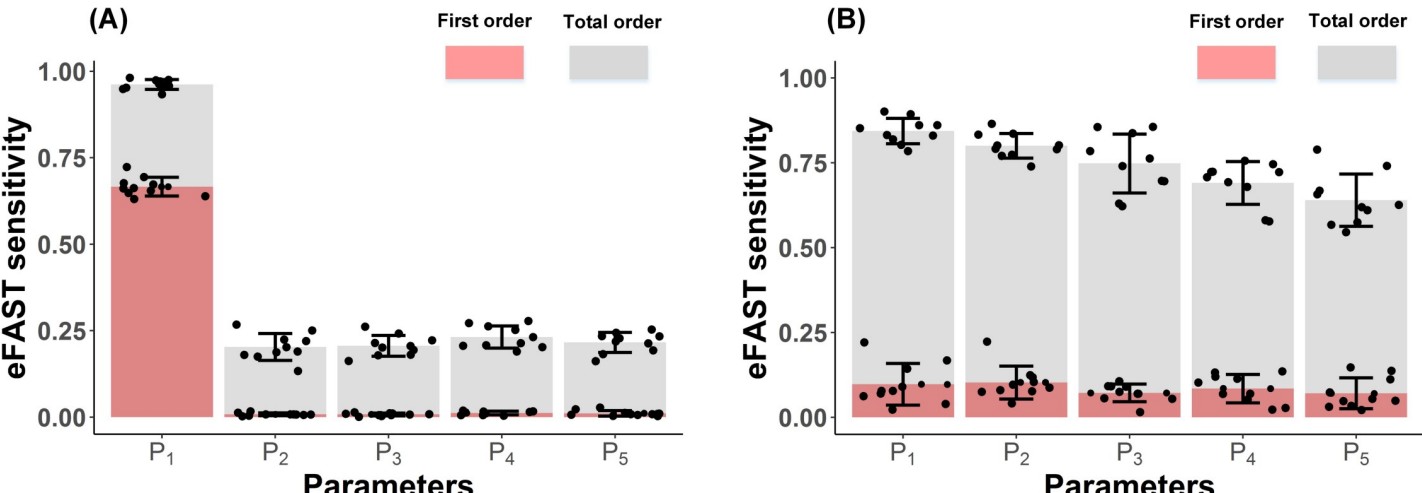

**Fig 7. Change in model output based upon selected parameters.** (A) eFAST sensitivity based upon full data set; (B) eFAST sensitivity based upon reduced data set that includes only cases with T1D occurrence. The horizontal axes represent five sensitive parameters. The parameters $P_1$, $P_2$, $P_3$, $P_4$, $P_5$ represent the average lifespan of cytotoxic $CD8^+$ T cells in islets, the initial number of damaged β cells ($\beta_{init}$ at simulation step 0), the time interval of DCs movement in islets, the maximum number of naïve $CD8^+$ T cell binding to DCs, and recruitment rate of DCs circulating in the pancreas, respectively. The vertical axes denote the eFAST sensitivity analysis for the first-order and total-order indexes. The first-order indexes are denoted by pink portions starting from 0, and total-order indexes are illustrated by both grey and pink portions. The first-order index reflects the variance induced by the uncertainty of a single input factor and the total-order index represents the output variance abiding by the interaction between this input factor and other input factors.

In **Fig 7B**, we observe that first-order and total-order effects of the five parameters changed drastically when reduced data sets (indicating only T1D incidences were considered for eFAST analysis) were included in the global sensitivity analysis. The purpose of implementing reduced data sets for global sensitivity analysis is to investigate how elevated these input factors perturb T1D progression and the onset of hyperglycemia or honeymoon phase when overt T1D was developed. Interestingly, we observed that the selected input factors have comparable main effects on variations of the model output (as shown in **Fig 7B**). Compared to full data sets, our findings show that more interaction effects on global sensitivity analysis of the model output were observed in the reduced data set. This indicates that interaction effects between the five parameters have more contribution to T1D progression once T1D occurrence is apparently inevitable.

## Incorporation of therapy-directed approaches

In the previous section, we applied sensitivity analysis methods to investigate the main and interaction effects of unknown parameters on T1D development. Five essential parameters were found to be highly correlated to overt T1D progression. Moreover, interaction effects were observed to be more related to the onset of T1D progression when T1D was developed. Considering the parameters (i) initial number of damaged β cells and (ii) average time interval of DCs movement in islets may be difficult to be regulated in experimental studies, we focused on the three parameters including the average lifespan of CTLs in islets, maximum number of naïve CD8$^+$ T cells binding to DCs, and recruitment rate of DCs for targeted therapeutic strategies. The objective of therapy-directed simulations is to test the following hypothesis: the occurrence of T1D is significantly delayed or inhibited by therapeutic interventions if: 1) the lifespan of CTLs in islets was reduced; 2) the capacity of naïve CD8$^+$ T cell binding to DCs was reduced; 3) the recruitment rate of DCs was reduced; or 4) two of the above strategies (1)-(3) were applied simultaneously by a combination of these therapeutic interventions.

As depicted in **Fig 8A**, the heatmap shows the probability of T1D occurrence was significantly declined if the average lifespan of CTLs in islets was reduced. Furthermore, the occurrence of T1D is dependent on the time administering therapeutic intervention (x-axis) and the applied amount of therapeutic dose range (y-axis). When the proposed therapy based upon our simulation findings was administered with a single dosage regiment at week 4, a significant decline in T1D occurrence was observed. However, when the same amount of therapeutic dose range was administered at a later stage of T1D progression (e.g. from week 14 to week

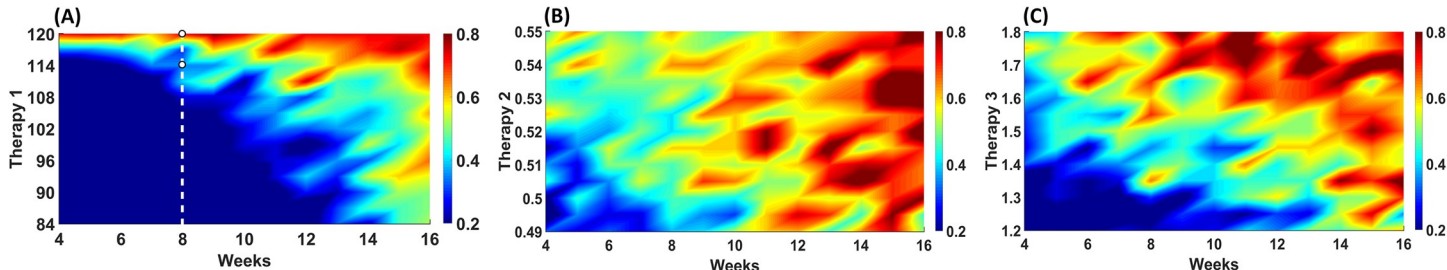

**Fig 8. Probability of developing T1D based upon different targeted therapeutic strategies.** The horizontal axes represent the administered time of a single dosage during simulation runs. Regions with red color represent a higher incidence of diabetes, and regions with blue color represent a lower incidence of diabetes. (A) Therapy 1 denotes the strategy that reduces the longevity of CTLs in islets. (B) Therapy 2 is described as an intervention that inhibits DC infiltration into islets. (C) Therapy 3 denotes a strategy that mitigates the binding process of naïve CD8$^+$ T cells on DCs. Therapeutic interventions were implemented based upon single dosage regimens starting at week 4 to week 16. Two black circles filled white in Panel (A) show the change in T1D occurrence in Therapy 1 when the lifespan of CTLs was reduced from 120 hours to 114 hours at week 8.

16), T1D occurrence is not significantly delayed or prohibited compared to an early stage of administering therapeutic intervention. Also, the percentage of T1D occurrence ranging from 60% to 70% is reduced to the range (30% to 40%), respectively, when the lifespan of CTL was reduced from 120 hours to 114 hours at week 8 (black circles filled white in **Fig 8A**). However, higher single-dose regimens or multiple-dose regimens are required to reduce T1D occurrence to 30% when therapeutic interventions are administered at week 12. Compared to targeted therapeutic strategies required to weaken the longevity of CTLs in islets, both therapies that reduce the capacity of naïve CD8[+] T cell binding to DCs and the recruitment rate of DCs in the pancreas are less effective, as illustrated in **Fig 5B** and **5C**.

The effects and efficacies of combinations of therapeutic strategies on T1D development were then investigated, as shown in **Fig 9**. Compared to a single therapeutic intervention, combinations of therapeutic strategies significantly reduce the likelihood of overt T1D instances at a late stage of T1D progression (week 12 as shown in **Fig 9** and week 14 as shown in **S5 Fig**). Specifically, the incidence of overt T1D approximately ranges from 50% to 70% when we implement a single therapy at week 12 (Therapy 2 or Therapy 3 as shown in **Fig 8B** and **8C**). However, the occurrence of overt T1D diminished and shifted the range (50% to 70%) to (20% to 40%) when a combination strategy was applied (as depicted in **Fig 9C1–9C3**).

These findings from therapy-directed simulations indicate that the effectiveness of the therapeutic strategies on T1D progression is dependent on three main factors: the time administering therapeutic interventions, the administered amount of therapeutic dose range, and the type of therapeutic intervention based upon the key parameters.

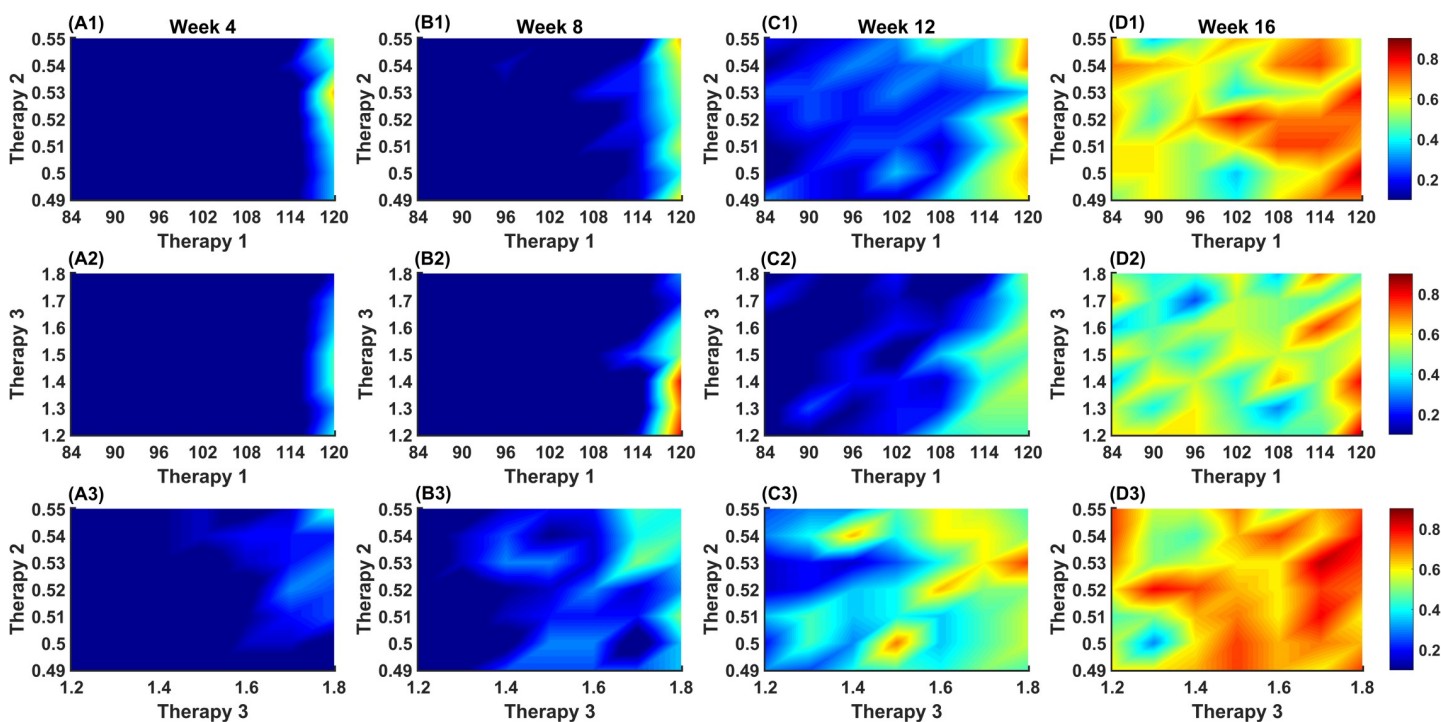

**Fig 9. Probability of developing T1D based upon combinations of proposed therapeutic strategies.** Therapeutic interventions were implemented based upon single dosage regimens starting at week 4 (Panels A1-A3) to week 16 (Panels D1-D3). Therapy 1 represents a certain strategy that reduces the residence of CTLs in islets. Therapy 2 is described as an intervention that can inhibit DC infiltration into islets. Therapy 3 denotes a strategy that prohibits binding sites on DCs for naïve CD8[+] T cells.

## Discussion

Human diseases frequently involve networks of complex inter- and intra-cellular and signaling interactions linking the immune system and metabolism during disease progression. Mathematical and computational approaches emerging from modeling the detailed biology of immune responses have been successfully used to explain non-intuitive behaviors and characterize variations in disease [79,80]. The development of T1D involves a complex network between pancreatic β cells and cells of both innate and adaptive immune systems [81], which requires a systematic level of understanding of onset, progression, and prevention of the disease. Herein, we sought to reconcile how cellular-level insights about the underlying interplay of immune responses and diabetes affect the ultimate behaviors of type 1 diabetes.

To accomplish this goal, we developed a hybrid modeling structure. Previous studies including our earlier research papers constructed mechanistic mathematical models to investigate various components in T1D and sepsis [4,9–12]. We have also developed data-driven ABM to carry out *in silico* therapy-directed experiments in a mouse model and to investigate immune responses in human cell lines [14, 82]. With previous experience and established knowledge in both mathematical modeling, ABM of immune responses, we identified the most accurate platform, as a promising framework for the physiological and etiological process to simulate complex cellular interactions in T1D, that integrates data-driven ABM and mathematical modeling with statistical components.

One major advantage of applying an agent-based framework to modeling a biological system is that it can capture spatial and noise-driven effects in highly orchestrated movements of different agents during a biological process that remains incompletely understood and is normally underestimated by mathematical models [15,42]. However, ABM, in some cases, would require extensive computational resources because of a detailed representation of the system. In realistic T1D progression, interactions among $10^6$–$10^7$ cells were observed [23,43–45]. In this case, when an agent-based simulation was implemented at such high computational complexity levels, the computational efficiency was relatively low (one simulation run took more than 10 hours to conduct a 20-week T1D progression using our validated model). Such high computational complexities would further hinder the sensitivity analyses because they require massive repetitions of simulation runs when varying parameter values [16]. To improve the computational efficiency and underperformance, we proportionally reduced the number of agents (each agent in-silico represents one cell in vitro, details are presented in the **Materials and Methods** section) in the agent-based simulation. The agent-based simulation with a reduced size can simulate a 20-week T1D progression within one hour using a high-performance computer workstation which allowed for further high-throughput computing analysis.

By implementing advanced statistical methods, we designed the hybrid computational framework to model the complex network which primarily focused on the simulation of interactions between DCs, cytotoxic CD8[+] T cells, and pancreatic β cells during T1D progression. Simulated results recaptured ($R^2$ = 0.98 based upon the comparison between mean values of model results and mean values of experimental results) individual trajectories of cytotoxic CD8[+] T cells in experiments, as illustrated in **Fig 4**, suggested that the model can calibrate and predict successfully the progression of the disease. Our simulations confirmed the occurrence of the individual-specific "honeymoon phase" by incorporating the combined effect of β-cell replication, functional recovery, and resistance to autoimmune destruction into the model. Owing to the hybrid framework, we could explain the complex, highly orchestrated, and robust physiology of cell fate and population heterogeneity when the underlying mechanisms are a noise-driven process. Despite the stochastic behavior of regulatory circuits within agents,

our results validated with experimental data show that cellular networks are precisely regulated leading to autoimmunity and beta-cells eventually succumb to damage (**Figs 4** and **5**).

The manifestation of overt T1D was shown to associate with the loss of β-cell mass [33,48,50]. Thus, we calibrated the kinetics of pancreatic β cells in our simulation runs and used the number of residual health β cells as a means to quantifying the magnitude of the initial stimulus for T1D progression. It was also shown that β-cell transplantation or regeneration is not sufficient for treating T1D disorder [83]. One major concern is that regenerated β cells may also become the new target of CTLs during T1D progression and β cells eventually succumb to damage and apoptosis [32] if the factors driving T1D progression fail to control homeostasis and immunological tolerance in alienating lymphocytes. To identify the driving factors leading T1D progression to overt T1D, we implemented local and global sensitivity analyses using *in silico* data generated in our simulations and suggested the average lifespan of cytotoxic CD8$^+$T-cells, initial number of apoptotic β-cells, number of binding sites on DCs for naïve CD8$^+$ T cells, time interval of DCs movement in islets, and recruitment rate of DCs as five key drivers of T1D progression. The time required for developing T1D was the primary output of interest in the sensitivity analyses, which was also the target of *in silico* therapy-directed experiments. Based upon these therapy-directed simulations, we discovered that the probability of T1D development could be reduced if appropriate strategies were applied at various time windows during T1D progression.

Cytotoxic CD8$^+$ T cell has long been recognized as a major driving force for developing T1D [84]. Several experimental studies discovered that the prevention of diabetes can be achieved in animal models by down-regulation of cytotoxic CD8$^+$ T cells [85–88]. Pinkse *et al.* [84] focused on administering immunodominant peptides derived from major antigens to down-regulate cytotoxic CD8$^+$ T cells associated with β cell destruction in type 1 diabetes. But due to a limitation of the peptide therapy (short half-life of peptides in the circulation), our simulating model results suggest that further studies on other therapeutic options to reduce the lifespan of cytotoxic CD8$^+$ T cells are needed to improve survival rates of beta cells. Several studies have provided a thorough discussion about multiple ways of regulating the longevity of immune cells including the use of agents, telomerase activation, inhibition of apoptosis, and reversal of energy [89–93]. These studies have demonstrated multiple methods can be employed to regulate the lifespan of immune cells, which provide additional justification to implement parameter modification such as the "lifespan of cytotoxic CD8$^+$ T cells" in our modeling. Besides, the model results demonstrated that the likelihood of T1D occurrence would significantly decline if the administered amounts of therapy were increased at a late stage of T1D progression (as shown in **Fig 8**), which could also be a potential research direction for future experimental studies.

Our simulations also suggested T1D progression would markedly be delayed if recruitment rates of DCs were declined (as depicted in **Fig 6E** and **6B**). Previous studies demonstrated a depletion of DCs helps preventing or reversing T1D in NOD mice [32,81,94]. Interestingly, our simulation study showed that this can be explained by the fact that, since the primary role of DCs is to function as APCs both in the periphery and within islets [32], the activation and the invasion of CTLs to islets decreased if the number of DCs was reduced in islets. As depicted in **Fig 8**, a reduction in the recruitment rate of DCs is less effective to improve survival at a late stage of T1D progression. However, it should be also noted that the computational model was focused on CD8$^+$ T cells only; thus, it is important to point out herein a certain level of mature dendritic cells may be also important for the generation of regulatory CD4$^+$ T cells of inhibiting T1D progression [95]. The reasons that CD4$^+$ T cells were not included in the model are discussed in-depth as follows.

Our simulations further suggested that the number of binding sites on DCs for naïve CD8+ T cells is another key parameter that could impact T1D progression. Early treatment of NOD mice was observed to reduce the likelihood of autoimmune diabetes incidence if the value of this parameter declined (as illustrated in **Fig 6D**). The binding process between naïve CD8+ T cells and DCs is a crucial step for the activation of naïve CD8+ T cells in PLNs [32,33,35]. Then the activated CD8+ T cells in PLNs migrate to islets and become cytotoxic CD8+ T cells. When the number of binding sites on DCs for naïve CD8+ T cells decreased, the window of opportunity for naïve CD8+ T cells to become activated CD8+ T cells narrowed, and therefore the number of cytotoxic CD8+ T cells in islets declined. Thus, the model suggests that a reduction of the capability of naïve CD8+ T cell binding to DCs would provide another opportunity and experimental direction for resolving T1D development. Fischer *et al.* [96] reported that naïve CD4+ T cells deprived of MHC class II molecules showed a decreased ability to interact with a limited number of cognate antigen-bearing dendritic cells. In their study, Act-mOVA mice were bred to I-Aβ-deficient (MHC class II-deficient) mice to produce Act-mOVA MHC class II-deficient mice. This evidence shows that it is highly likely that a similar method could be applied to a NOD mouse model to deprive receptor molecules on dendritic cells with *in vivo* knockout method, which may limit the binding sites on dendritic cells to interact with naïve CD8+ T cells. As a matter of fact, this study provided the empirical investigation of our primary intuition for the concept of our simulation.

Moreover, results from therapy-directed simulations suggest that a combination of therapeutic strategy on reducing both recruitment rate of DCs and the maximum number of naïve CD8+ T cells binding to DCs would noticeably reverse the mechanism of destruction of beta cells, a potential to cure overt T1D, from week 10 to week 12 (as plotted in the heatmaps of **Fig 9** and **S5 Fig**). Thus, unlike single-drug interventions, the effect of combined therapeutic strategies (e.g. a combination of treatment strategy on reducing both recruitment rate of DCs and number of binding sites on DCs for naïve CD8+ T cells) on T1D treatment may provide valuable insight into a prospective cross-sectional as well as longitudinal studies of onset, remission, and recurrence. Therapy-directed simulations recommend that the combined therapeutic strategies as an important mediator would contribute to a decreased risk of a range of short-term and long-term diabetes-related complications. It is most noteworthy that a recent study [97] also demonstrated combination therapy can reverse hyperglycemia in a NOD mouse model with established type 1 diabetes.

The present study should be evaluated in the context of several possible limitations. First and foremost, like all models, the hybrid computational model presented herein is an abstract of reality, to some extent, similar to an experiment when it is performed for a laboratory investigation dealing with a controlled environment while it might give a completely different output in reality and the human body. Although the model was designed to describe the major components based on the current knowledge and available data, excluded multiple mechanisms associated with T1D progression should be mentioned herein. For example, we did not model CD4+ T cells in this study because the role of CD4+ T cells in autoimmune T1D remains incompletely understood [35] and thus experimental data on CD4+ T cells are not sufficient. It is worthwhile to mention that the progression of T1D is associated with the finely tuned immune balance between effector CD4+ T cells and regulatory CD4+ T cells. The quantitative relationship between regulatory CD4+ T cells and effector CD4+ T cells is regulated by other cell types and cytokines [32,35,98–100]. In this case, additional interactions between dendritic cells, CD4+ T cells, and CD8+ T cells would require extensive efforts in estimating unknown parameters and their distributions. When the number of unknown parameters is very large, it is often impossible to identify reliable model results [101]. A second limitation concerns the discrepancy between the NOD mouse model and the human model. Despite the NOD mouse

model has been successful in multiple forms for studying T1D in humans [23], the limitation of the NOD mouse model should be marked. Compared to the NOD mouse model, it seems that the human pancreas has a lower potential for β-cell regeneration based on recent access to the pancreas of organ donors with T1D [102]. In addition, the severity of insulitis in human models has been found to be less pronounced than in the NOD mice model [103], which may impact the conclusions that parameters associated with CD8[+] T cells predominate T1D progression in human models. Based on these reasons, it is worth mentioning that the strategies that would be effective in the NOD mouse model are not necessarily effective in clinical trials. We also considered modeling other treatment parameters such as anti-CD3 because they were proven to be effective in NOD mice and clinical trials [104]. However, modeling these treatment parameters, again, required the incorporation of numerous unknown parameters at the current stage.

Despite these limitations, our study enhanced the feasibility of computational modeling for simulating disease progression by simulating complex cellular networks. Biological research has been long focused on various aspects of the mystery of biological systems, which advanced the understanding of mechanistic knowledge of complex systems. Nevertheless, many of the emergent, integrative behaviors of biological systems result not only from complex interactions within a specific level but also from feedback interactions that comprise complex cellular networks [80]. This hybrid computational approach can simulate complex T1D progression associated with higher rates of diabetes-related complications by capturing essential components and their interactions during multiple pathways of immune responses. It can also investigate the effects of parameters on T1D development and suggest future directions for experimental studies to reduce these complications. With the aid of advanced statistics, our findings uncovered non-intuitive biological parameters that could be potentially targeted as therapeutic options. Accordingly, we suggest that this hybrid computational framework can help improve the systematic understanding of complex diseases and design *in silico* therapeutic strategies for other complex diseases such as cancer.

## Supporting information

**S1 Table. Agent types and shapes in agent-based modeling (ABM).**
(DOCX)

**S2 Table. Simulation initialization and agent set up in ABM environment.**
(DOCX)

**S3 Table. Agent rules during T1D progression.** Rules were described using pseudocode in NetLogo, and state variables associated with agents were identified by square brackets).
(DOCX)

**S4 Table. Unknown parameters in the ABM simulations.** Parameter ranges used for local sensitivity analysis and Latin hypercube sampling (LHS) are tabulated below. By consulting with the field experts and cited papers, the ranges were found in the literature (in this case literature does are not necessarily mean about NOD mouse model since these parameters are defined as unknown) which were then employed for unknown parameters (i.e. Ranges used for LHS) for the purpose of initial simulation of the model. The ranges determined by the field experts are marked using asterisks.
(DOCX)

**S5 Table. Computational costs for sensitivity analysis and therapy-directed simulations.**
(DOCX)

**S1 Fig. A binding process between naïve CD8$^+$ T cells and dendritic cells in the ABM simulations.** During the binding process, each APC (i.e. dendritic cell that has engulfed antigens) checked the surrounding eight patches. If naïve CD8$^+$T cells appear on these eight patches, an APC checks the number of dendritic cells binding to each naïve CD8$^+$T cell. A dendritic cell can bind to a naïve CD8$^+$T cell if this naïve CD8$^+$T cell has available binding sites for the dendritic cell (e.g. the naïve CD8$^+$ T cell at the bottom of nine grids in **S1 Fig**). A dendritic cell can bind to multiple naïve CD8$^+$ T cells until there are no available binding sites on the naïve CD8$^+$ T cell (e.g. the naïve CD8$^+$ T cell in the top of nine grids in **S1 Fig**). The number of binding sites on dendritic cells for naïve CD8$^+$ T cells are determined by a state variable named the *maximum number of naïve CD8$^+$ T cells binding to DCs.*
(TIFF)

**S2 Fig. Transformation from the search curve to sampling points in a two-dimensional space.** (A): Search curves defined by Eq (8). The solid line denotes a search curve with a frequency equal to 24, and the dashed line represents the search curve with a frequency equal to 1. (B): Scatterplot of the sampling points in a two-dimensional space. Each point in the space represents a combination of two factors ($q_1$ and $q_2$). (C): CDF of a normal distribution that transforms the sampling points from a unit space [0,1] to the realistic values (represented by $X_i$).
(TIFF)

**S3 Fig. Data samples of 5 parameters drawn from normal distributions based upon eFAST.** The horizontal axis represents the total sample size ($n = 1000$) for five parameters, and the vertical axis represents parameter values.
(TIFF)

**S4 Fig. Non-sensitive parameters based on local sensitivity analysis.** Discernible fluctuation in time required for developing overt T1D based upon selected values of parameters using local sensitivity analysis. The horizontal axis represents selected values of five input parameters in the spectrum of $[P_d \pm 30\% \times P_d]$, and the vertical axis denotes the time required for developing T1D. Box plots and raw data illustrate heterogeneities occurring during T1D for selected values. Black dotted lines outline lower and upper whiskers and black solid boxes show $Q_1$, $Q_2$, $Q_3$ for first quartile, interquartile, and third quantile values for selected points within the spectrum of $[P_d \pm 30\% \times P_d]$. For non-sensitive parameters, the $p$-values of $F$ tests were greater than the predetermined significance level (e.g. $\alpha = 0.05$).
(TIFF)

**S5 Fig. Therapy-directed simulations Probability of developing T1D based on a combination of proposed therapeutic strategies.** Therapy interventions were implemented based upon single dosage regimens starting at week 6 to week 14. Regions with red color represent a higher incidence of diabetes, and regions with blue color represent a lower incidence of diabetes. Therapy 1 represents a certain strategy that reduces the residence of CTLs in islets. Therapy 2 is described as an intervention that can inhibit DC infiltration into islets. Therapy 3 denotes a strategy that prohibits binding sites on DCs for naïve CD8$^+$ T cells.
(TIFF)

**S1 Movie. NetLogo interface during T1D progression.** The cross-section captures two-dimensional projection along the largest axis of the pancreas. This window, in fact, simulates part of the whole picture. For illustrative purposes, we show a few islets and agents during T1D progression. The window is partitioned into two panels; the top panel shows PLNs with circulation in a large red disk, and the bottom panel illustrates the islets including beta cells in green

colors. Brown agents represent cytotoxic CD8+T cells, yellow for autoantigens, pink agents for activated CD8+T cells, and green agents turning black shaded areas show damaged and apoptotic beta-cells. Insulitis in NOD mice proceeds through a stage of peri-insulitis in which T cells accumulate around the islets. For improving the computational efficiency, the video only shows insulitis progressing from inside the islets.
(MOV)

## Acknowledgments

The authors gratefully acknowledge Dr. Christophe Benoist from Harvard Medical School for kindly sharing with us the raw data of their experimental study. The authors also would like to thank Mal Rooks Hoover, CMI for helping create Figs 1 and 2.

## Author Contributions

**Conceptualization:** Zhenzhen Shi, Majid Jaberi-Douraki.

**Formal analysis:** Zhenzhen Shi, Majid Jaberi-Douraki.

**Funding acquisition:** Majid Jaberi-Douraki.

**Methodology:** Zhenzhen Shi, Majid Jaberi-Douraki.

**Project administration:** Majid Jaberi-Douraki.

**Resources:** Majid Jaberi-Douraki.

**Software:** Zhenzhen Shi, Majid Jaberi-Douraki.

**Supervision:** Majid Jaberi-Douraki.

**Validation:** Zhenzhen Shi, Yang Li, Majid Jaberi-Douraki.

**Visualization:** Zhenzhen Shi, Majid Jaberi-Douraki.

**Writing – original draft:** Zhenzhen Shi, Majid Jaberi-Douraki.

**Writing – review & editing:** Zhenzhen Shi, Yang Li, Majid Jaberi-Douraki.

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
