## [Decision Letter · Decision Letter 0]

23 Apr 2021

Dear Dr. Shi,

Thank you very much for submitting your manuscript "Hybrid computational modeling connecting data science and statistical learning deciphers the expedience of simulating complex cellular networks in type 1 diabetes" for consideration at PLOS Computational Biology.

As with all papers reviewed by the journal, your manuscript was reviewed by members of the editorial board and by several independent reviewers. In light of the reviews (below this email), we would like to invite the resubmission of a significantly-revised version that takes into account the reviewers' comments. In particular, both reviewers requested clarification about the role of AI/machine learning approaches in the development of the model and connecting the simulation results to experimental data. An example of confusing language is provided in the title itself, which mentions connecting "data science" and "statistical learning" -- my understanding is that statistical learning is part of data science. Also the phrase "deciphers the expediency" doesn't have a very clear meaning. I think the authors are trying to say "demonstrates the utility of."

We cannot make any decision about publication until we have seen the revised manuscript and your response to the reviewers' comments. Your revised manuscript is also likely to be sent to reviewers for further evaluation.

Sincerely,

James R. Faeder

Associate Editor

PLOS Computational Biology

Mark Alber

Deputy Editor

PLOS Computational Biology

Reviewer's Responses to Questions

**Comments to the Authors:**

Reviewer #1: Shi and co-authors present a study aimed at integrating data-driven and mechanistic (agent-based) modeling in the context of the immune dysregulation underlying Type 1 Diabetes (T1D). Overall, I am concerned that the authors conflate the data-driven and statistical methods used in the analysis of model output with a true integration of data-driven and mechanistic modeling. I would argue that to truly integrate these modeling approaches, the authors would have had to use machine learning approaches to help define model rules and/or parameters, rather than simply using those tools to analyze model output. I am also concerned that no true, prospective validation of key model predictions has been carried out. Thus, it is unclear to me that such a nominally complex architecture was needed in order to perform the studies presented in the manuscript. My specific concerns are as follows:

1. Introduction: Some of the statements made by the authors are paradoxical. For example, they state that “…the barrier in sampling experimental data from pancreatic tissues or lymph nodes in high-risk T1D subjects makes the use of quantitative modeling approaches based on AI machine learning of pancreatic β-cell destruction an intriguing opportunity to analyze this disease.” It would seem that the exact opposite is true: the absence of extensive time course data would appear to make data-driven modeling/machine learning very difficult. They also refer to “mathematical data science models” and “data-oriented agent-based modeling,” which to me is an oxymoron: mechanistic mathematical and agent-based models are based on biological principles derived from the aggregate knowledge gleaned from the literature, rather than on specific datasets (though, of course, mechanistic models can be calibrated to data). In contrast, data-driven/machine learning models are largely based on data rather than on principles. The authors need to read through the paper carefully and make sure that such statements, which make it appear as if the authors are not familiar with these core concepts, are modified or removed.

2. Introduction: the authors mostly cite their own prior work on modeling T1D. Even a cursory search of PubMed lists a substantial number of mechanistic models of T1D in addition to the one paper cited by Linderman et al, and these studies should be acknowledged and discussed. Likewise, there have been review/perspective articles that have called for the integration of mechanistic and data-driven modeling approaches, and some of those papers should be cited as well.

3. Introduction: the paragraph which introduces the NOD mouse model is jumbled and should be revised. As written, it is difficult to tell if the authors are saying that obtaining data in this model is useful or not, as they appear to argue against themselves.

4. Introduction (line 123): the authors do not give an adequate rationale as to why a hybrid model is the best approach. As mentioned in point #1 above, the authors argue that data are insufficient, and in point #3 they make a weak case for obtaining data in the NOD mouse model. Based on that, I do not see how “…based on the above facts, (they) proposed a novel hybrid agent-based model by integrating phase progressions in complex pancreatic cellular networks, mathematical modeling, biological parameters in the domain of machine learning models, ABM, and statistical learning components into a computational framework to simulate T1D progression using experimental data and clinical evidence from NOD mouse models.” I would argue that the exact opposite is true, based on those same facts. Ultimately, the authors have chosen to take this approach, but the “facts” as given are not sufficient support for why they chose to do so.

5. Results: the authors jump right to validation of their model against data. They need to first describe the model at least in high level (abstracting what is in the Methods) and give key performance metrics of the model (e.g. ability to simulate important divergent outcomes associated with progression, or lack thereof, to T1D). This brings up another important point: are the authors simulating a state of existing T1D, or a progression to T1D (or not, depending on simulation conditions) that then can be queried and perturbed? The latter would be preferable to the former, because simulating a situation where the investigators simply posit that “this is T1D” means that they have fully addressed the dynamics of the biological processes that lead up to that point. The statements further down suggest that the model is in fact one of T1D progression, but this needs to be made explicit and supported with details of the model. Furthermore, it is unclear what role machine learning played in the generation of this model. Was a data-driven modeling analysis performed in order to help define model rules, key interactions, feedbacks, or was machine learning only used in the analysis of the parameterization and output of the model? If the latter, then this is really not novel as this type of analysis of output of mechanistic models has now become fairly standard.

6. Results, Fig. 1: what was the number of experimental repeats (animals/samples)? Also, the authors state that “…a significant increase in infiltrating CD8+ T cells occurs around week 8 in

the simulation.” How do they define “significant”? Was a statistical analysis carried out? Also, I am concerned that the relatively tight confidence intervals in the data have resulted in an overfit model (see my point #7 below).

7. How does the Latin Hypercube Sampling approach to defining parameter values handle the effect of stochasticity in the ABM? How were these parameter choices validated? How is overfitting avoided? In the clinical setting of T1D, I would expect much larger confidence intervals around the dynamics of cells and inflammatory mediators, and hence parameter choices that appear to work in this experimental setting may not be useful in truly understanding T1D.

8. Results, Fig. 2: As per point #6, the authors again state that “…counts of healthy β cells start to

9. significantly decrease at around week 8.” Again, how was significance determined?

10. Results: The authors state that “…(they) applied sensitivity analysis methods to investigate the main and interaction effects of unknown parameters on T1D development, and their domain of machine learning.” What does “their domain of machine learning” mean? This is a key point, since I am still unclear as to what role machine learning has played in these analyses (see my point #5 above).

11. Figs. 5 & 6: no details are given as to what “statistical learning” was carried out (and why).

Reviewer #2: The manuscript reports the development of a hybrid computational modeling based on agent-based-modeling (ABM) to simulate some aspects of the complex cellular networks in type 1 diabetes (T1D). A comprehensive effort is made to investigate the development of Type 1 diabetes based on the behavior of some components of the immune system. The model is validated by data from NOD mice. The contributions and clarity of the manuscript can be enhanced by addressing the issues listed below.

The structure of the agent-based model (the structure of the agents, the layers, Net-Logo) should be presented earlier in the text to inform the readers about the characteristics and limitations of the ABM developed. One of the agents should also be given to illustrate how the agents developed are interacting and the level of “intelligence” of the agents – how they perceive their environment, how the decide on their action, and how these actions are implemented.

The components of the immune system included in the model and the components left out that may have an effect on the progress should be described clearly. All interactions of various activities of the immune system that lead to T1D may be too complex to fit in a Net-Logo model or even to develop their rules. But it would be valuable to the readers to know the limitations of the model and the phenomena that it describes/simulates.

An exploratory sensitivity analysis is made by varying one input at a time. Of course, this does not permit capturing the effects of interactions between the inputs that may have significant effects on the outputs. A cautionary note to warn the reader would be useful.

Fourier amplitude sensitivity test (eFAST) method is used for global sensitivity analysis. Some brief remarks on why this method is preferred over Sobol sensitivity analysis should be included in the text.

The destruction of islets and beta cells by using ABM was reported by Xu, Ozturk, and Cinar in some recent papers: Agent-Based Modeling of Immune Response to Study the Effects of Regulatory T Cells in Type 1 Diabetes. Processes, 2018; and Agent-Based Modeling of the interaction between CD8+ T cells and Beta cells in Type 1 Diabetes, PLoS ONE, 2018. Their ABM is also based on mouse models. These should be included in the references and some of their results should be compared with the results of the ABM model reported in this manuscript.

The terms AI, machine learning, and AI machine learning are used in the manuscript several times. The ABM is a system with distributed AI, and this should be mentioned when the ABM is introduced. The use of machine learning should be pointed out and the term “AI machine learning“ should be clarified or removed. Machine learning encompasses the capture of underlying relations or behavior in data by processing the data. Most of the results reported are the results of simulations. The results are reported by using statistical analysis techniques such as ANOVA. If there are any automated classifications (for example by using support vector machines, neural networks, discriminant analysis) the term machine learning would be appropriate and pointed out in the manuscript.

**Have the authors made all data and (if applicable) computational code underlying the findings in their manuscript fully available?**

Reviewer #1: Yes

Reviewer #2: Yes

PLOS authors have the option to publish the peer review history of their article (what does this mean?). If published, this will include your full peer review and any attached files.

Reviewer #1: No

Reviewer #2: No
---

## [Decision Letter · Decision Letter 1]

1 Sep 2021

Dear Dr. Shi,

We are pleased to inform you that your manuscript 'Hybrid computational modeling demonstrates the utility of simulating complex cellular networks in type 1 diabetes' has been provisionally accepted for publication in PLOS Computational Biology.

Best regards,

James R. Faeder

Associate Editor

PLOS Computational Biology

Mark Alber

Deputy Editor

PLOS Computational Biology

Reviewer's Responses to Questions

**Comments to the Authors:**

Reviewer #1: The authors have addressed my core critiques.

Reviewer #2: The authors have addressed the comments of the reviewer. Another spelling check is necessary to catch and correct some typos.

**Have the authors made all data and (if applicable) computational code underlying the findings in their manuscript fully available?**

Reviewer #1: Yes

Reviewer #2: Yes

PLOS authors have the option to publish the peer review history of their article (what does this mean?). If published, this will include your full peer review and any attached files.

Reviewer #1: No

Reviewer #2: No

---

## [Editor Report · Acceptance letter]

20 Sep 2021

PCOMPBIOL-D-21-00374R1 

Hybrid computational modeling demonstrates the utility of simulating complex cellular networks in type 1 diabetes

Dear Dr Shi,

I am pleased to inform you that your manuscript has been formally accepted for publication in PLOS Computational Biology. Your manuscript is now with our production department and you will be notified of the publication date in due course.

With kind regards,

Andrea Szabo
